# SFX-01 is therapeutic against myeloproliferative disorders caused by activating mutations in Shp2

Hyun-Ju Cho[1,8], Joy Smith[1,2,8], Christopher H Switzer[3,8], Eleni Louka [4], Rebecca L Charles [1], Oleksandra Prysyazhna [1], Ewald Schroder[2], Mariana Fernandez-Caggiano[1], Daniel Simoes de Jesus [1], Seda Eminaga[2], Xiaoke Yin [5], Xiaoping Yang[6], Steven Lynham[6], Manuel Mayr[5], Valle Morales [6], Katiuscia Bianchi [6], Vinothini Rajeeve [7], Pedro R Cutillas [7], Adam J Mead[4] & Philip Eaton [1✉]

## Abstract

Activating mutations of Src homology-2 domain-containing protein tyrosine phosphatase-2 (Shp2) cause multiple childhood conditions for which there is an unmet therapeutic need, including juvenile myelomonocytic leukemia (JMML) and Noonan syndrome. SFX-01, an α-cyclodextrin-stabilized sulforaphane complex currently in clinical development, covalently adducts cysteine residues. Using unbiased proteomics, its protein targets were identified, including Shp2. SFX-01 induced an inhibitory dithiolethione modification at the Shp2 active site cysteine. Importantly, in a transgenic mouse model of human Noonan syndrome with hyperactive D61G Shp2, SFX-01 concomitantly normalized their phosphatase activity and myeloid cell count. Furthermore, SFX-01 also attenuated JMML human patient-derived hematopoietic stem cell proliferation that was linked to STAT1 signaling and decreased cyclin D1 expression, resulting in cell-cycle arrest. We conclude that SFX-01 is an activating mutant Shp2 inhibitor and may offer beneficial effects in patients with JMML or Noonan syndrome.

**Keywords** Shp2; SFX-01; Sulforaphane; Noonan Syndrome; Myeloproliferative Disorders
**Subject Categories** Cell Cycle; Genetics, Gene Therapy & Genetic Disease; Pharmacology & Drug Discovery

## Introduction

Src homology domain 2 (SH2)-containing protein-tyrosine phosphatase-2 (Shp2) regulates downstream signaling events such as cellular proliferation and cell-cycle progression by activating Ras while inhibiting STAT1 signaling (Matozaki et al, 2009; You et al, 1999). Activating mutations of Shp2, which drive oncogenic Ras signaling, are associated with human cancers and Noonan syndrome (NS) (Tartaglia et al, 2001; Xu et al, 2010; Zhang et al, 2015). Furthermore, activating Shp2 mutations result in myeloid cell proliferation, as observed in NS and juvenile myelomonocytic leukemia (JMML) patients (Xu et al, 2010). Mutations in Ptpn11 that code for Shp2 are not only the most frequent cause of JMML, but these patients have the worst prognosis (Miao et al, 2020). Although small-molecule allosteric inhibitors of Shp2, such as SHP099, have been developed, these agents do not significantly limit the activity of mutant Shp2 (Chen et al, 2016; LaRochelle et al, 2018; Raveendra-Panickar et al, 2022; Sun et al, 2018; Yuan et al, 2020). Thus, there are currently no drugs approved for JMML, and the only option is bone marrow transplantation with a success rate of only ~50% (Gupta et al, 2021); therefore, a molecular entity that inhibits mutant Shp2 is urgently needed.

Sulforaphane (SFN) is an isothiocyanate compound derived from cruciferous vegetables and is associated with increased cellular antioxidant capacity and anti-inflammatory effects. SFN reacts with protein cysteine residues to alter their function and thus modulate cellular homeostasis. Although SFN or preparations containing it have shown therapeutic promise against human cancers in clinical trials (Bauman et al, 2022; Yuan et al, 2025), its instability hampers its utility and regulatory approval. This issue has been addressed by the development of SFX-01, a synthetic form of D, L-sulforaphane stabilized as an α-cyclodextrin complex. The active sulforaphane component has a molecular mass and physiochemical properties that fits each of Lipinski's rule-of-five (Lipinski, 2004), indicating it has druglike properties. Indeed, multiple clinical safety or efficacy trials have been performed with SFX-01 (clinicaltrials.gov IDs NCT01948362, NCT02055716, NCT02970682, NCT02614742) (Clack et al, 2025; Howell et al, 2019; Long et al, 2024; Zolnourian et al, 2024; Zolnourian et al, 2020). It is evident that SFX-01 has shown clinical safety in healthy and unwell humans, and its notable that it is effective in patients with metastatic breast cancer (Howell et al, 2019).

[1]William Harvey Research Institute, Faculty of Medicine and Dentistry, Queen Mary University of London, London, UK. [2]King's College London British Heart Foundation Centre, School of Cardiovascular and Metabolic Medicine & Sciences, London, UK. [3]Department of Molecular and Cell Biology, University of Leicester, Leicester, UK. [4]Medical Research Council (MRC) Molecular Haematology Unit, MRC Weatherall Institute of Molecular Medicine, National Institute for Health Research Biomedical Research Centre, University of Oxford, Oxford, UK. [5]National Heart and Lung Institute, Imperial College London, London, UK. [6]King's College London, Proteomics Facility, Centre of Excellence for Mass Spectrometry, London, UK. [7]Barts Cancer Institute, Queen Mary University of London, London, UK. [8]These authors contributed equally: Hyun-Ju Cho, Joy Smith, Christopher H Switzer. ✉E-mail: p.eaton@qmul.ac.uk

The therapeutic action of SFN has been attributed to its ability to induce phase 2 enzymes and increase antioxidant levels by adduction to KEAP1 and activation of the KEAP1/Nrf2 system (Hu et al, 2011; Kensler et al, 2013; Zhang et al, 1992). However, increasing cellular reducing status in cancer patients by administering antioxidant compounds has been deleterious (Alpha-Tocopherol, 1994), and chronic Nrf2 activity is correlated with poor prognosis in some cancers (Lignitto et al, 2019; Zhao et al, 2020). Furthermore, SFN is anticipated to be an indiscriminate thiol electrophile. Therefore, alternative, and perhaps multiple molecular targets are likely to mediate the observed antiproliferative effects of SFN and SFX-01.

In this study, Shp2 was identified as an SFX-01 molecular target from an unbiased proteomic analysis utilizing an antibody developed to detect in vivo SFN-modified proteins. Functional assays indicated that SFN-modification inhibited its phosphatase activity via covalent dithiolethione modification. In a transgenic mouse line expressing an activating mutant Shp2, SFX-01 restored the myeloid cell number to wild-type levels and resulted in cell cycle arrest via hyperactive STAT1 signaling. Therefore, SFX-01 is identified as an in vivo mutant Shp2 inhibitor and may have therapeutic effects for JMML and other Shp2-driven pathologies.

# Results

## Shp2 is a molecular target of SFX-01 in cells and in mice

Human embryonic kidney (HEK) 293 cells were exposed to either R-, S-, or R/S-SFN and multiple SFN-modified proteins were detected by immunoblotting and immunofluorescence staining (Fig. 1A,B). The cysteinyl-SFN antibody is specific to protein-SFN covalent modification as (1) the antibody is not immunoreactive to tissue unless exposed to SFN, and (2) SFN-conjugated bovine serum albumin (BSA), which functions as a neutralizing peptide, blocks the immunoreactivity of the cysteinyl-SFN-specific antibody (Appendix Fig. S1). Wild-type (WT) mice receiving an oral gavage of either SFN or SFX-01, a-cyclodextrin encapsulated R/S-SFN, revealed numerous SFN-adducted proteins in multiple tissues compared to vehicle-treated mice (Fig. 1C). SFN-adducted proteins immunoprecipitated from cardiac tissue (Fig. 1D) were identified using LC-MS/MS (Table EV1). In this way, Shp2 was found to be a reproducible in vivo target of SFN (Fig. 1E).

## Sulforaphane inhibits Shp2 activity via covalent modification

WT recombinant human Shp2 activity was inhibited in a concentration- and time-dependent manner when reacted with SFX-01 (Fig. 2A,B). SFX-01 also inhibited Shp2 co-treated with bisphosphorylated insulin receptor substrate 1 (IRS1) (Fig. 2C). Incubation of Shp2 with excess SFX-01 resulted in SFN-adduction detected by immunoblotting (Fig. 2D). Shp2 showed a time-dependent loss of cysteinyl-SFN antibody immunodetection when reacted with lower SFX-01 concentrations despite phosphatase activity being maximally inhibited (Fig. 2B).

LC-MS/MS analysis revealed a dithiolethione moiety bridging cysteine residues 333 and 367 induced by SFX-01 (Fig. 1E; Appendix Fig. S2). Cysteines 333 and 367 are proximal to the

catalytic Cys-459. SFN likely reacts via an initial adduction to form a thiocarbamate intermediate, then a second cysteine attacks to produce the dithiolethione (Fig. 2E,F). SFN-induced dithiolethione formation was previously shown using a model dithiol-containing compound (Zhang et al, 1996), however identifying its formation in a protein highlights its likely biological importance.

To investigate the stability and transformation of the SFN adduct, HEK 293 cells were transfected with WT, C333/367S, or C333/459S Shp2 double mutants and exposed to SFX-01. The WT Shp2 showed transient SFN-labeling that diminished over time, consistent with dithiolethione formation, while the double cysteine mutants retained SFN labeling at 4 h post-treatment (Fig. 2G). These data support the concept that the auxiliary cysteines are required for the formation of a dithiolethione and thus for the time-dependent loss of the SFN signal. Furthermore, phosphatase activity was retained in the auxiliary cysteine mutants but abolished in the active site mutant, indicating the requirement of the catalytic cysteine for enzymatic function (Appendix Fig. S3).

## SFX-01 inhibits mutant Shp2 activity in a transgenic mouse model

SFX-01 inhibited hyperactive Shp2 in $Ptpn11^{D61G(-/+)}$ mice (Fig. 3A,B). Protein-SFN adducts accumulated over time in cardiac and liver tissues (Appendix Fig. S4), and SFX-01 was stable in the drinking water over the 10-day treatment period (Appendix Fig. S5). Shp2 inhibition in vivo was not associated with detectable SFN adduction (Fig. 3C), likely due to continuous low-dose exposure, in contrast to the acute bolus dosing used in the proteomics screen.

To directly assess the redox status of Shp2 and investigate dithiolethione formation, immunoprecipitated Shp2 from cardiac tissue of treated or untreated mice was incubated with biotin-iodoacetamide, which covalently binds to free thiol groups. Reduced biotinylation in SFX-01-treated samples indicated that Shp2 thiols were oxidized, consistent with dithiolethione formation (Fig. 3D). To further confirm oxidation of vicinal thiol groups, samples were labeled with biotin-phenylarsenic acid (PAA), which selectively binds pairs of closely positioned thiol groups. Shp2 from untreated WT and Ptpn11$^{D61G(-/+)}$ mice showed strong biotin-PAA labeling, whereas Shp2 from SFX-01-treated mice displayed a significantly reduced signal (Fig. 3E), consistent with cysteine oxidation and dithiolethione bridge formation that disrupts vicinal thiol recognition. Together, these experiments demonstrate that SFX-01 modifies Shp2 in vivo via dithiolethione formation, inhibiting its phosphatase activity in both WT and mutant mice.

## Testing SFX-01 toxicity in Noonan syndrome model mice

Mice carrying the $Ptpn11^{D61G(-/+)}$ mutation are commonly used to model NS, as this mutation phenocopies many aspects of the human syndrome, including short stature and myeloproliferative disease (Kratz et al, 2005; Romano et al, 2010). While the mutant Shp2-mediated musculoskeletal defects are patterned in utero, as expanded on below, myeloproliferative neoplasms develop during adulthood. Consequently, it was logical to investigate the ability of SFX-01 to limit the myeloproliferative disorder present in the $Ptpn11^{D61G(-/+)}$ mice. However, before embarking on experiments assessing this, we conducted toxicity studies with WT and $Ptpn11^{D61G(-/+)}$ mice treated with or without SFX-01 for 10 weeks. SFX-01 significantly accelerated growth in WT mice, resulting in

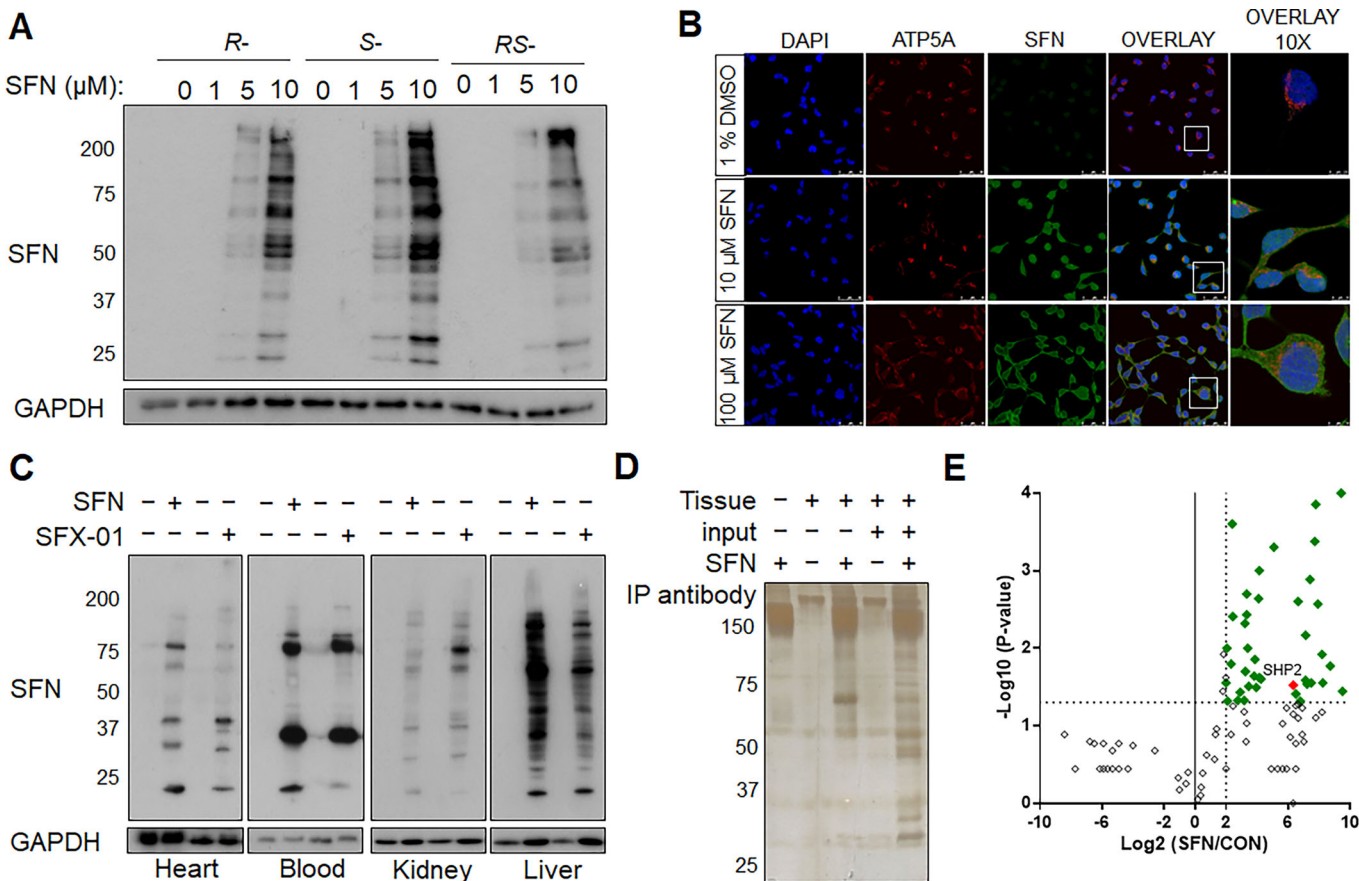

**Figure 1.  Detection and identification of sulforaphane-modified proteins.**

(A, B) Immunoblot and immunofluorescence analysis showing multiple proteins in HEK 293 cells modified in a concentration-dependent manner by SFN upon treatment with R-, S- or R, S-isomers. (C) Immunoblot analysis showing multiple proteins modified by SFN in widespread tissues 3-h after oral gavage of WT mice with SFN or SFX-01. (D) A silver-stained polyacrylamide gel showing immunoprecipitation of SFN-adducted proteins from the cardiac tissue of WT mice using the anti-SFN antibody following oral gavage of the electrophile. (E) Volcano plot showing cardiac proteins in the upper right quadrant that were significantly and reproducibly modified by SFN in mice ($n = 4$ biological replicates). Source data are available online for this figure.

markedly increased body weight compared to vehicle-treated controls ($P < 0.0001$). However, the drug had no effect on growth in heterozygous $Ptpn11^{D61G(-/+)}$ mice. SFX-01 increased food intake on average by 2.5 g per WT mouse per week compared to control WT mice ($P = 0.0051$), and this ~10% increase in calories likely explains the additional weight gain (Appendix Fig. S6). Food intake by the $Ptpn11^{D61G(-/+)}$ mice was not altered by the presence of SFX-01, consistent with the drug not changing their growth rate. Comprehensive plasma biochemistry analyses did not identify changes induced by SFX-01. Specifically, no evidence of gastro-intestinal tract, cardiac, renal or liver damage was observed in either genotype following SFX-01 treatment (Appendix Fig. S6B,C). Although these results are consistent with studies showing SFX-01 is well tolerated, including in humans (Clack et al, 2025), importantly we found that this was also the case in NS model mice.

## Chronic SFX-01 treatment reduces myeloid hyper-proliferation

Blood leukocyte populations were compared in 12-week-old WT or $Ptpn11^{D61G(-/+)}$ mice and again after 10 weeks of vehicle or SFX-01

treatment using Wright-Giemsa staining. While exposure to vehicle did not alter their leukocyte count in WT mice, the $Ptpn11^{D61G(-/+)}$ transgenics on this control intervention showed a significant increase in the number of these cells (Fig. 4A). These observations reflect progression of NS pathology in $Ptpn11^{D61G(-/+)}$ mice over time as anticipated. Notably, SFX-01 treatment significantly reduced leukocyte populations in both WT and $Ptpn11^{D61G(-/+)}$ mice (Fig. 4A), mirroring the reduction in Shp2 activity observed in mice exposed to the drug (Fig. 3A,B). Importantly, SFX-01 reduced $Ptpn11^{D61G(-/+)}$ mouse leukocytes to levels comparable to those observed in vehicle-treated WTs (Fig. 4B). Leukocyte populations were further analyzed for myeloid-derived cells (CD11b+/Ly6G+ and CD11b+/Ly6C+) by flow cytometry (Appendix Fig. S7). $Ptpn11^{D61G(-/+)}$ mice had significantly more myeloid cells in their blood compared to WT mice exposed to vehicle (Fig. 4C), consistent with the histological and Shp2 activity analyses. SFX-01 treatment reduced myeloid cell populations in $Ptpn11^{D61G(-/+)}$ mice to similar levels observed in vehicle-treated WT mice. As Shp2 is implicated in the differentiation of white blood cell progenitors to the myeloid cell lineage (Jack et al, 2009), these observations are rationally explained by SFX-01 attenuating myeloid cell

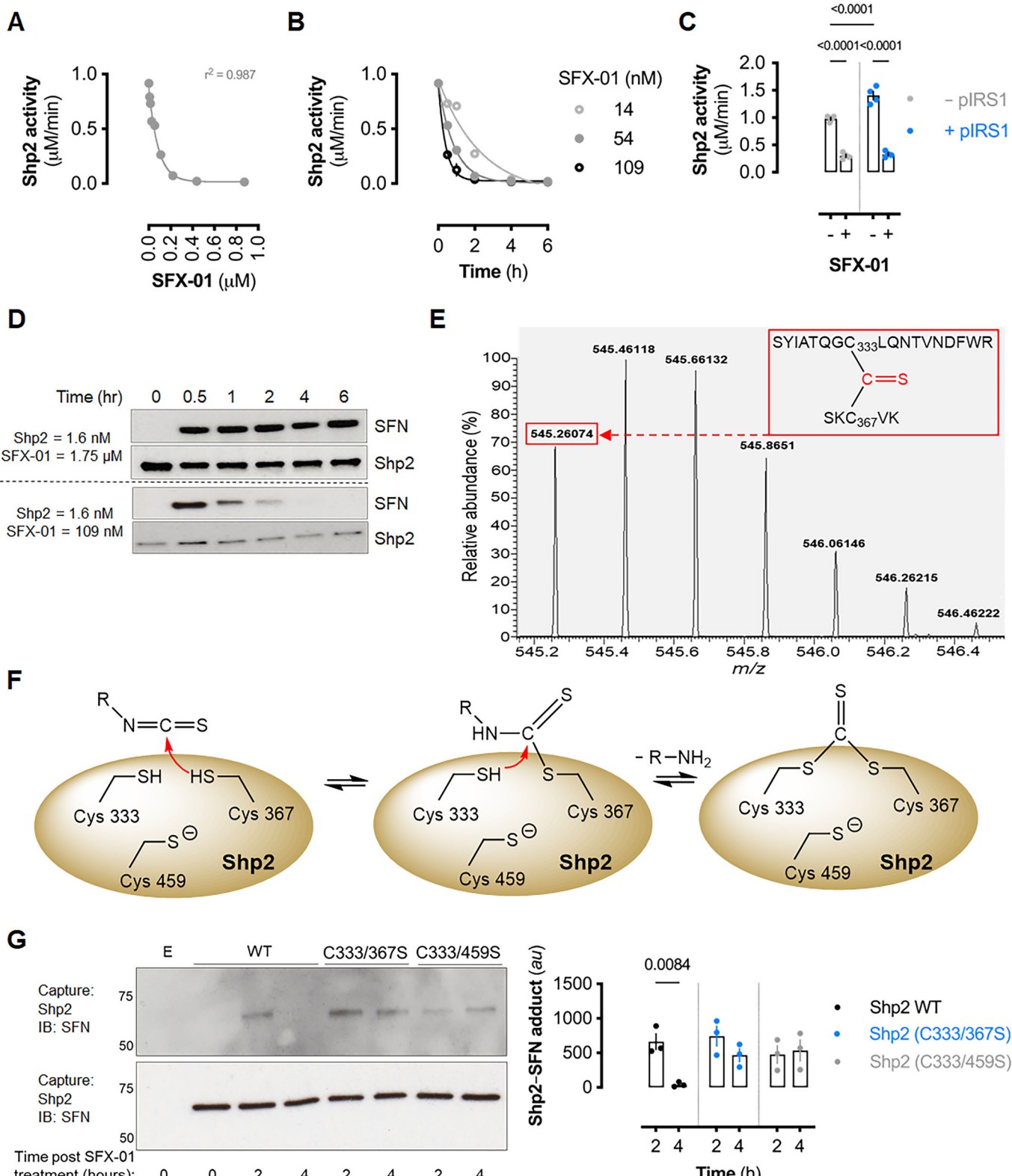

**Figure 2. Sulforaphane adducts Shp2 to induce an inhibitory dithiolethione modification in vitro.**

(A) Recombinant human Shp2 activity after 30-min incubation with SFX-01. Data represent mean activity (± SEM; $n = 3$ biological replicates) and were fitted to a one-phase exponential decay curve (gray line; $r^2 = 0.987$). (B) Shp2 activity after incubation with SFX-01 for the indicated times and concentrations. Data shown are mean activity (± SEM; $n = 3$ biological replicates). (C) Shp2 activity after 30-minute incubation with or without bisphosphorylated IRS1 and SFX-01. Bar represents mean activity (± SEM; $n = 4$ biological replicates) and $P$ values calculated by two-way ANOVA with Sîdak post hoc test. (D) Representative immunoblots showing SFN-modification of recombinant Shp2 (1.6 nM) following incubation with 1.75 or 0.109 µM SFX-01 for 30 min. (E) Precursor isotopic envelop spectrum of 0.1 µM recombinant human Shp2 protein incubated with equimolar SFN for 6 h at 37 °C corresponding to a dithiolethione modification adducted between $Cys^{333}$ and $Cys^{367}$. (F) Schematic representing the proposed mechanism of Shp2-dithiolethione formation by SFN. The isothiocyanate group reacts with a cysteine residue to form a dithiocarbamate intermediate, which further reacts with a second cysteine residue to yield the dithiolethione modification. (G) Immunoblot of immunoprecipitated WT or active site mutant Shp2 and SFN-modification from HEK cells treated with SFX-01. "E" represents non-transfected cells, and 0 h represents untreated cells. The graph represents densitometric analysis of Shp2-SFN adduct formation in WT or mutant Shp2 exposed to SFX-01 for 2 or 4 h. Bars represent mean values (± SEM; $n = 3$ biological replicates) and $P$ values calculated by two-way ANOVA with Sîdak post hoc test. Source data are available online for this figure.

proliferation by inhibiting the activity of the mutant Shp2 that otherwise mediates the aberrant growth.

$Ptpn11^{D61G(-/+)}$ mice exhibit splenomegaly because of their myeloproliferative disease (Fig. 4D; Appendix Fig. S8) (Araki et al, 2004). Spleen sizes were measured by ultrasound in each genotype before and after 10 weeks of exposure to vehicle or SFX-01. Although mice exposed to vehicle for 10 weeks showed an increase in spleen size regardless of their genotype (Fig. 4D), administration of SFX-01 only decreased spleen size in the $Ptpn11^{D61G(-/+)}$ transgenics and not in WTs (Fig. 4E). Like the observations made in peripheral blood, SFX-01 also caused a significant decrease in $Ptpn11^{D61G(-/+)}$ mouse spleen myeloid cell populations (Fig. 4F).

Whether SFX-01 was able to rescue other pathogenic characteristics present in $Ptpn11^{D61G(-/+)}$ mice is a rational consideration. These transgenics demonstrate many characteristic phenotypic features of NS, including cardiomyopathy (Yi et al, 2016) and defective platelet aggregation (Bellio et al, 2019), as well as cardiac valve defects and musculoskeletal abnormalities likely primarily patterned in utero. Indeed, the homozygous D61G mutant is embryonic lethal, and the heterozygotes have reduced viability, presumably because of developmental defects (Araki et al, 2004). To determine if SFX-01 might correct this, we conducted studies in which pregnant dams were treated with SFX-01 to assess whether it enhanced the birth rate of heterozygous offspring or enabled homozygous D61G mutant offspring to survive to birth, which to reiterate does not otherwise occur. SFX-01 exposure during development not only failed to produce viable homozygous D61G mutant offspring but also led to the complete loss of heterozygous $Ptpn11^{D61G(-/+)}$ mice, which are normally viable. This outcome was consistent across two separate studies. In the first, SFX-01 was administered from pre-conception through pregnancy and in the second it was given only post-gastrulation (Appendix Fig. S9). The reason for this unexpected outcome remains unclear but may relate to phospho-activation of ERK that occurs in mice or cells exposed to SFX-01 observed both by immunoblotting for pERK and with unbiased phosphoproteomics analysis (Appendix Fig. S13D–F) (Krenz et al, 2008; Krenz et al, 2005; Nakamura et al, 2007). Importantly, we found pERK levels in newly delivered neonates from WT / WT crosses in which the mother was exposed to SFX-01 were markedly increased compared to vehicle controls (Appendix Fig. S9I). It is difficult to understand why administering SFX-01 to mothers carrying offspring from HET/HET crosses did not deliver any pups. This complete loss of offspring only occurred at the 2.5 mg/ml SFX-01 concentration (Appendix Fig. S9D), whereas

some WT offspring were delivered when the lower 0.8 mg/ml concentration was used (Appendix Fig. S9H).

## SFX-01 induces cell cycle arrest via increased STAT1 signaling

Shp2 phosphatase activity negatively regulates STAT1 signaling to induce cellular proliferation (You et al, 1999). Consistent with decreased myeloid proliferation and attenuated Shp2 activity upon exposure to SFX-01, this treatment also significantly increased STAT1 phosphorylation in leukocytes of mice regardless of their genotype (Fig. 5A). SFX-01 decreased cyclin D1 expression in leukocytes of WT or $Ptpn11^{D61G(-/+)}$ mice (Fig. 5A), which is explained by SFX-01 inducing STAT1 signaling that is coupled to cyclin D1 protein degradation and cell cycle arrest (Dimco et al, 2010; Masamha and Benbrook, 2009). SFX-01 also induced STAT1 phosphorylation as well as loss of cyclin D1 expression in the human acute myelomonocytic leukemia GDM-1 cell line that expresses an activating Shp2 mutation associated with JMML and NS (Fig. 5B). In addition to cyclin D1 degradation, STAT1 activation results in protein kinase R (PKR) phosphorylation, which limits growth by attenuating protein synthesis (Handy and Patel, 2013; Wong et al, 1997). SFX-01 induced PKR phosphorylation in both WT and $Ptpn11^{D61G(-/+)}$ leukocytes (Appendix Fig. S10A), further indicating that STAT1 signaling was induced by SFX-01-mediated Shp2 inhibition. SFX-01 resulted in a concentration-dependent increase in the percentage of GDM-1 cells in $G_0/G_1$ phase, indicating a $G_1$ cell cycle arrest (Fig. 5C), that is rationally explained by the decreased cyclin D1 expression. SFX-01 also concentration-dependently decreased GDM-1 colony formation (Fig. 5D). Furthermore, moderate concentrations of SFX-01 resulted in an increased $G_0/G_1$ cell population in mutant Shp2-expressing JMML patient-derived hematopoietic stem cells (HSCs) compared to those isolated from control cord blood (CB) (Fig. 5E). At higher concentration, SFX-01 induced more JMML HSCs into the sub-$G_0$ phase compared to those isolated from CB control cells. Consistent with cell cycle arrest, 50 or 100 µM SFX-01 significantly reduced JMML-derived HSC colony formation compared to control cells (Fig. 5F). Therefore, SFX-01 not only limits myeloid cell proliferation by inducing STAT1 phosphorylation but also limits mutant Shp-2-driven JMML patient-derived HSC proliferation, highlighting the therapeutic potential of SFX-01. Whilst the enhanced susceptibility of JMML patient-derived versus control CB cells to growth inhibition is an important finding, this only manifests significantly at 50 µM SFX-01. This perhaps relates to experimental design in which SFX-01 was administered once at the start of a 14-day protocol, and the time-averaged concentration is likely lower. Indeed,

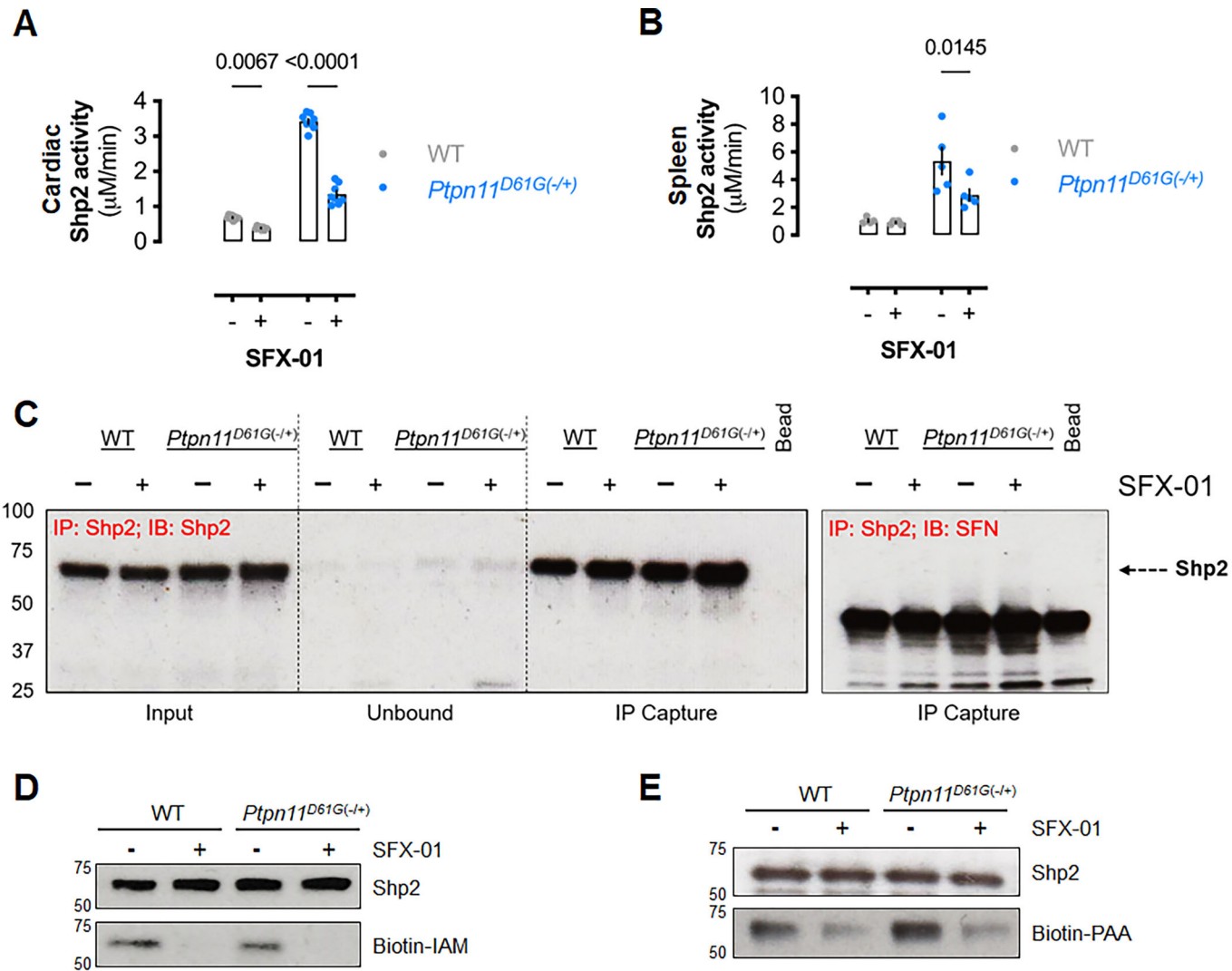

**Figure 3. SFX-01 inhibits wild-type and mutant Shp2 phosphatase activity in vivo.**

Shp2 phosphatase activity in (A) cardiac tissue ($n = 8$ mice) and (B) spleen of WT ($n = 4$ mice) or $Ptpn11^{D61G(-/+)}$ mice ($n = 5$ mice) that received SFX-01 in the drinking water for 10 days. (A, B) Bars represent mean activity (± SEM) and $P$ values calculated by two-way ANOVA with Sîdak post hoc test. (C) Representative immunoblots showing the input and immunoprecipitated Shp2 protein and SFN-labeled Shp2 from cardiac tissue isolated from mice with or without oral SFX-01 administration as in (A). (D) Representative immunoblot of Shp2 immunoprecipitated from cardiac tissue of WT mice following treatment with SFX-01 for 4 days after incubation with biotin-iodoacetamide. (E) Representative far-western immunoblot of Shp2 immunoprecipitated from cardiac tissue of WT mice following treatment with SFX-01 for 4 days. The immunoblot was then exposed to biotin-phenylarsenic acid (PAA). Source data are available online for this figure.

given the ability of sulforaphane to adduct to proteins, we indexed its bioavailability when it was added to the JMML cell culture media and found it decreased significantly over time (Appendix Fig. S10B). Unfortunately, the scarcity of these patient-derived cells, which readily differentiate to lose their phenotype with passaging, severely hampers further investigations.

## Insights to signaling induced by SFX-01 from phosphoproteomics

We performed phosphoproteomics to determine in an unbiased manner whether SFX-01 altered protein phosphorylation in CD11b$^+$ myeloid bone marrow cells isolated from WT or $Ptpn11^{D61G(-/+)}$ mice. It was notable that SFX-01 significantly

decreased the phosphorylation of Shp2 Y580 in CD11b$^+$ myeloid bone marrow cells from $Ptpn11^{D61G(-/+)}$ mice (Fig. 5G). Shp2 Y580 phosphorylation increases activity of the phosphatase (Lu et al, 2001), and thus the decrease in occupancy of this site is consistent with SFX-01 being inhibitory. Decreased Shp2 phosphorylation at Y580 is rational because its phosphatase activity regulates its own phosphorylation state. Shp2 self-regulates through auto-dephosphorylation, including at Y580. As Y580 phosphorylation can stabilize the closed and inactive conformation, this can attenuate the activity of upstream kinases that promote Y580 phosphorylation. This may cause a reduction in phosphorylation of Y580 because Shp2 does not maintain the feedback mechanisms necessary for its activation, culminating in a decrease in its own phosphorylation and overall activity. Consistent with SFX-01

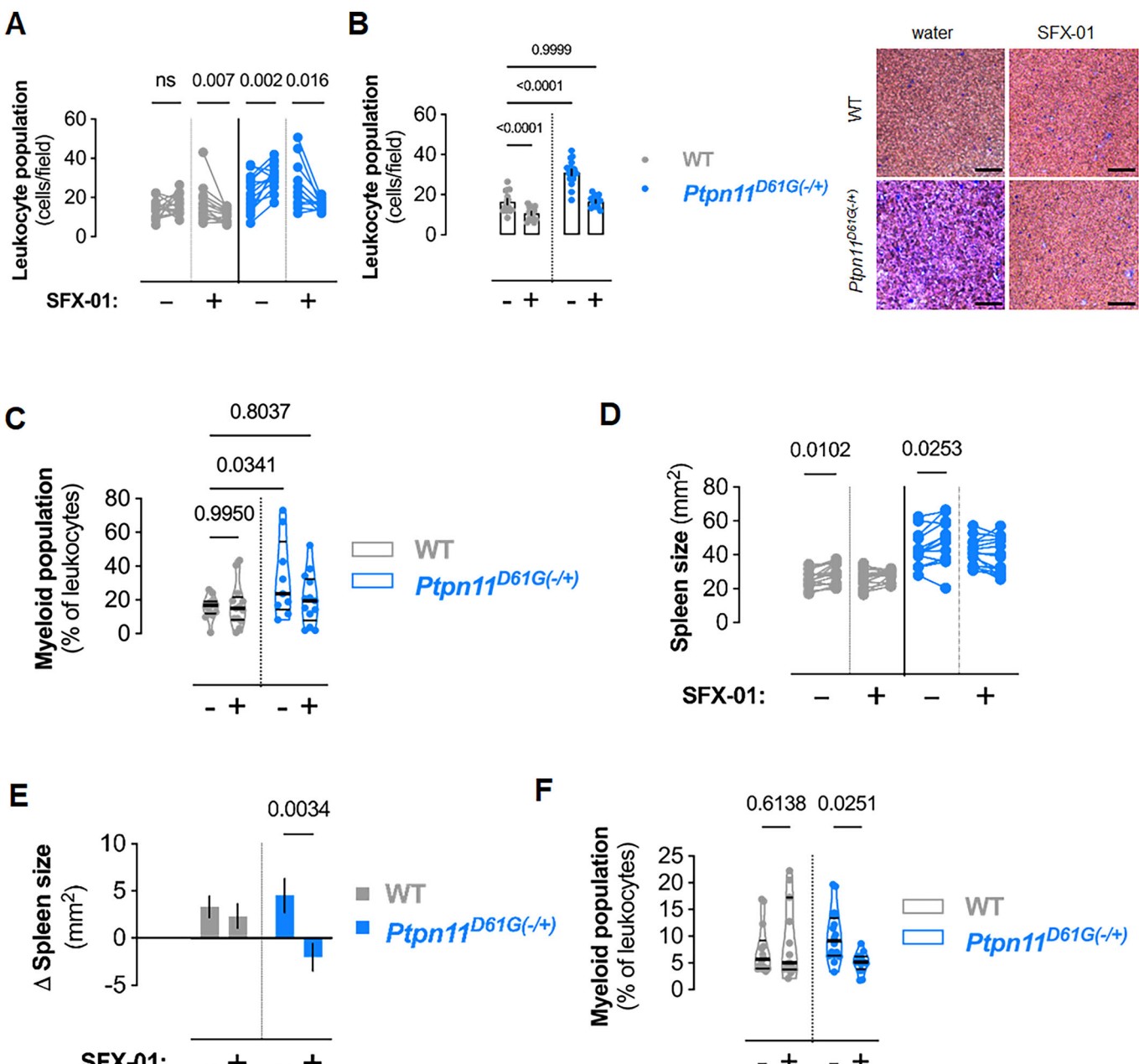

inhibiting Shp2 in CD11b+ cells from *Ptpn11*^D61G(−/+) mice, multiple established substrates of this phosphatase demonstrated increased phosphorylation upon exposure to the drug (Fig. 5G).

We reasoned that if SFX-01 inhibits Shp2 that there should be evidence of its substrates increasing their phosphorylation status, which occurs as shown in Fig. 5G. Pathway enrichment analysis was performed and stratified by those that contain Shp2 substrates or interactions, as shown in Appendix Fig. S11. It was evident that pathways that relate to growth, proliferation and cell cycle were negatively regulated when SFX-01 was present (Fig. 5). There are some caveats, however, that warrant further consideration of the findings from the phosphoproteomics analysis. For example, the heatmap in Fig. 5G shows proteins that are known interactors or

substrates of Shp2, but many of the phosphorylations measured were at serine or threonine residues. Although there is evidence for Shp2 dephosphorylating phospho-threonine (Kai et al, 2010), this phosphatase is thought to predominantly target protein phospho-tyrosine substrates, and so phosphorylation changes on other residues are likely because of events secondary or independent of the phosphatase being inhibited. Further complexity arises from SFX-01 having anti-proliferative actions that are not Shp2 dependent, as discussed below. If such pathways are engaged by SFX-01, it is feasible that they initiate signal transduction that affects Shp2 and its substrates, given their role in cell proliferation control, meaning there may be overlapping patterns of phosphorylation that arise from alterations to several connected pathways.

◄ **Figure 4. SFX-01 is therapeutic against myeloid disease in Noonan syndrome mice.**

(A) Changes in leukocyte cell populations in blood isolated from individual WT or $Ptpn11^{D61G(-/+)}$ mice before and after 10 weeks of SFX-01 in drinking water. $P$ values were calculated by paired t tests. For WT mice: $n = 14$ before and after vehicle treatment, $n = 14$ before SFX-01 treatment, and $n = 15$ after SFX-01 treatment. For $Ptpn11^{D61G(-/+)}$ mice: $n = 14$ before vehicle treatment, $n = 16$ after vehicle treatment, $n = 11$ before SFX-01 treatment, and $n = 14$ after SFX-01 treatment. (B) Blood leukocyte cell population after 10 weeks SFX-01 treatment. (*left*) Bars represent mean leukocyte population ( ± SEM) and $P$ values calculated by two-way ANOVA with Sîdak post hoc test. For WT mice: $n = 14$ vehicle treatment, and $n = 15$ SFX-01 treatment. For $Ptpn11^{D61G(-/+)}$ mice: $n = 16$ vehicle treatment, and $n = 14$ SFX-01 treatment. (*right*) Representative microphotographs of Wright-Giemsa-stained blood isolated from mice after 10 weeks SFX-01 treatment or controls. Scale bars show 200 μm. (C) Violin plots of blood myeloid-derived (CD11b$^+$/Ly6G$^+$ and CD11b$^+$/Ly6C$^+$) cell populations from WT or $Ptpn11^{D61G(-/+)}$ mice after 10 weeks of SFX-01 in drinking water. Horizontal lines represent quartiles and $P$ values calculated by two-way ANOVA with Sîdak post hoc test. For WT mice: $n = 13$ vehicle or SFX-01 treatment. For $Ptpn11^{D61G(-/+)}$ mice: $n = 9$ vehicle treatment, and $n = 13$ SFX-01 treatment. (D) Changes in spleen size from individual WT or $Ptpn11^{D61G(-/+)}$ mice before and after 10 weeks of SFX-01 in drinking water. $P$ values were calculated by paired $t$ tests. For WT mice: $n = 14$ before and after vehicle treatment ($P = 0.0102$), $n = 14$ before and after SFX-01 treatment ($P = 0.0890$). For $Ptpn11^{D61G(-/+)}$ mice: $n = 13$ before and after vehicle treatment ($P = 0.0253$), $n = 15$ before and after SFX-01 treatment ($P = 0.1694$) for each group. (E) Mean changes in spleen size in WT or $Ptpn11^{D61G(-/+)}$ mice over 10 weeks of SFX-01 treatment compared to before treatment. Bars represent mean spleen size ( ± SEM) and $P$ values calculated by two-way ANOVA with Sîdak post hoc test. For WT mice: $n = 14$ vehicle or SFX-01 treatment. For $Ptpn11^{D61G(-/+)}$ mice: $n = 13$ vehicle treatment, and $n = 15$ SFX-01 treatment. (F) Violin plots of spleen myeloid-derived (CD11b$^+$/Ly6G$^+$ and CD11b$^+$/Ly6C$^+$) cell populations from WT or $Ptpn11^{D61G(-/+)}$ mice after 10 weeks of SFX-01 in drinking water. Horizontal lines represent quartiles and $P$ values calculated by two-way ANOVA with Sîdak post hoc test. For WT mice: $n = 14$ vehicle treatment, and $n = 15$ SFX-01 treatment ($P = 0.6138$). For $Ptpn11^{D61G(-/+)}$ mice: $n = 16$ vehicle treatment, and $n = 12$ SFX-01 treatment ($P = 0.0251$). Source data are available online for this figure.

## SFX-01 is not fully selective for Shp2

SFX-01 inhibits hyperactive mutant Shp2 and ameliorates the myeloproliferation present in $Ptpn11^{D61G(-/+)}$ mice and cells from JMML patients. However, SFX-01 interacts with many molecular targets in vivo (Table EV1), and the extent to which its therapeutic effects are mediated by Shp2 inhibition was explored by comparing SFX-01 with SHP099. We conducted mitochondrial function tests using a Seahorse Analyzer to ascertain the impact, if any, of SFX-01 on mitochondrial function. Three or 10 μM SFX-10 or SHP099 did not significantly alter oxygen consumption rate (OCR) (Appendix Fig. S12). These findings indicate that SFX-01 are unlikely to exert its therapeutic actions by altering mitochondrial function. SHP099, a small-molecule Shp2 inhibitor, was reported as highly selective for this phosphatase because expression of mutant active-Ras rescued the growth impairment otherwise observed when Shp2 expression was genetically decreased (Chen et al, 2016). To compare the selectivity of SFX-01 or SHP099 for Shp2, HEK 293 cells expressing either active Shp2 (D61A) or active KRas (G12C) were individually treated with each drug and the impact on cellular proliferation was measured over 96 h. As expected, expression of either mutant Shp2 or active-KRas increased growth rates compared to mock-transfected cells (Appendix Fig. S13A,B). However, SFX-01 or SHP099 attenuated cell proliferation regardless of the expression of active Shp2 or active KRas. While this appears inconsistent with Chen et al (Chen et al, 2016), it is notable that they did not formally show the attenuated proliferation caused by SHP099 was rescued by expression of active Ras. Indeed, we note studies that demonstrated some mutant KRas cell models are sensitive to SHP099 (Mainardi et al, 2018), especially when grown as 3D multicellular preparations (Hao et al, 2019). Indeed, KRas-driven proliferation can be dependent on Shp2 (Mainardi et al, 2018; Ruess et al, 2018), and so there is complexity in definitively proving that either of these drugs are solely through inhibition of this phosphatase. In addition, we showed that the reduction in proliferation achieved individually by SFX-01 or SHP099 is not enhanced when both interventions are combined. This finding supports the conclusion that SFX-01 attenuates proliferation significantly through its inhibitory action on Shp2. If alternative mechanisms played a significant role, an additive effect would be expected when the SHP099 inhibitor was also present - an outcome that was not observed (Fig. EV1). Despite SFX-01 attenuating cell growth, including that stimulated by either mutant Shp2 or active-KRas, this was unexpectedly associated with increased pERK (Appendix Fig. S13C). Consistent with this, we observed an increase in pERK in mice exposed to SFX-01 (Appendix Fig. S13D). Furthermore, phosphoproteomics analysis showed that treatment with SFX-01 increased mitogen-activated protein kinase (MAPK) 1 or 3 activity and enriched the MAPK signaling pathway in WT and $Ptpn11^{D61G(-/+)}$ CD11b$^+$ cells (Appendix Fig. S13C–F). This is likely explained by an off-target effect, for example, phosphatases such as DUSP/MKP that dephosphorylate pERK have reactive catalytic cysteines that are conceivably inhibited by SFX-01. It is notable that the HSCs from one of the JMML patients (ID23) contain mutations in both Shp2 and Ras, but their hyper-proliferation was comparably attenuated by SFX-01 to those with mutations solely in the phosphatase. It is also likely, based on the phosphoproteomics analysis (Fig. 5G), that SFX-01 inhibits Shp1, which is not unexpected given its homology with Shp2.

## Discussion

Shp2 phosphatase activity couples to multiple signaling pathways that regulate cell growth, proliferation and differentiation (Qu and Feng, 1998; You et al, 1999), and this role is underscored by its involvement in both normal physiological and oncogenic signaling. Gain-of-function mutations in Shp2, particularly those that reside within the N- or C-terminal SH2 domains that disrupt the closed, auto-inhibited conformation of the phosphatase, have been implicated in various cancers as well as in developmental disorders including NS and JMML (Bentires-Alj et al, 2004; Chan et al, 2009; Mohi and Neel, 2007). Pharmacological efforts to inhibit hyperactive Shp2 have led to the development of allosteric inhibitors such as SHP099, which selectively stabilizes the inactive conformation of the phosphatase and effectively limit the proliferation of cancer cell lines driven by Ras/Erk pathway mutations (Fedele et al, 2018). However, these compounds are largely ineffective against Shp2 mutants that adopt an open conformation (Chen et al, 2016; LaRochelle et al, 2018; Sun et al, 2018).

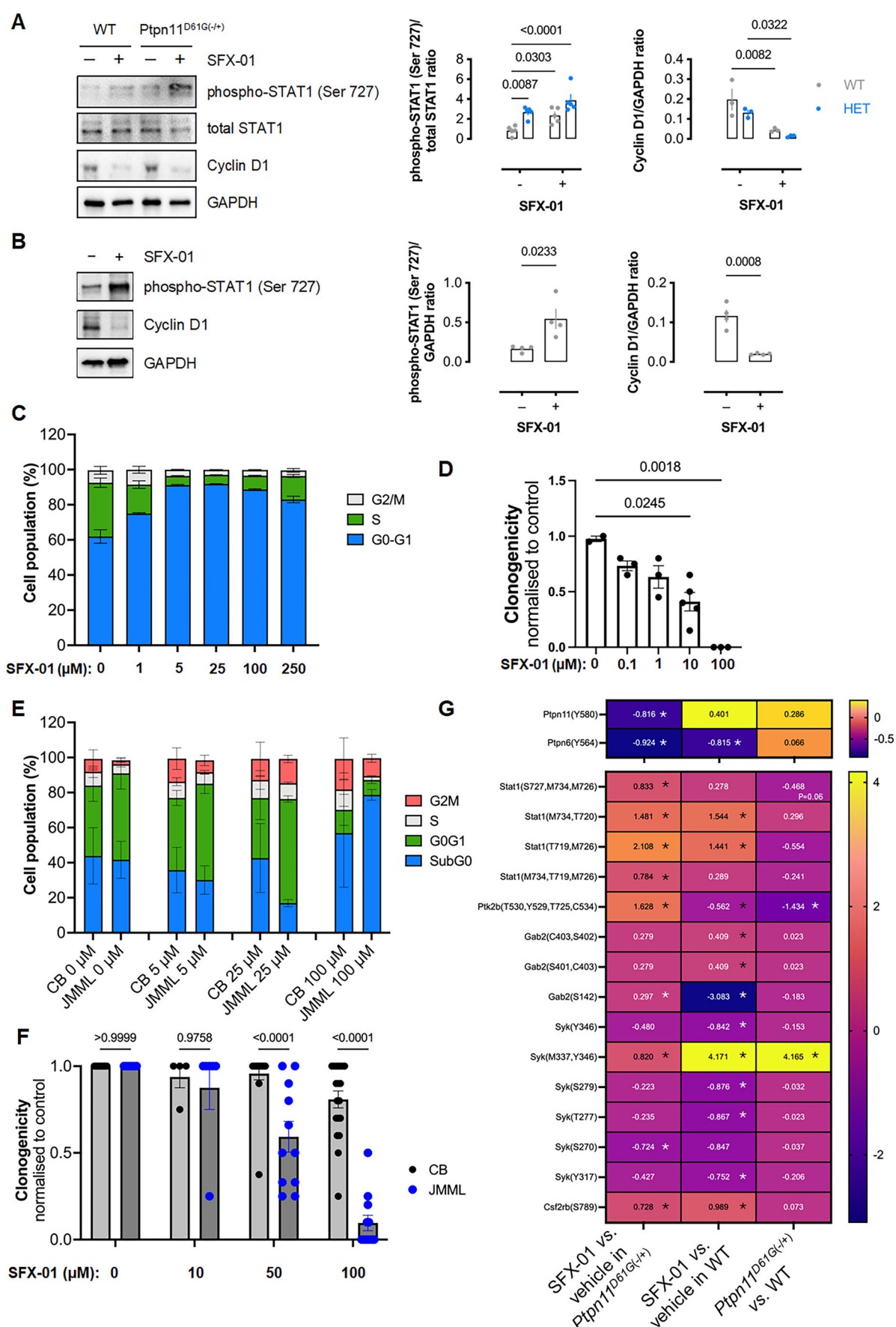

**Figure 5. SFX-01 induces cell cycle arrest via hyperactive STAT1 signaling.**

(A) Representative immunoblot of relative STAT1 phosphorylation and cyclin D1 expression in leukocytes isolated from WT or Ptpn11$^{D61G(-/+)}$ mice after 10-week SFX-01 treatment. Graphs represent densitometric analyses of relative STAT1 (Ser 727) phosphorylation and cyclin D1 expression. Bars represent mean expression level ( ± SEM) and P values calculated by two-way ANOVA with Sîdak post hoc test. $n = 5$ mice for STAT1 (Ser 727) phosphorylation and $n = 3$ mice biological replicates for cyclin D1 expression. (B) Representative immunoblots of relative STAT1 phosphorylation and cyclin D1 expression in a Shp2-activating mutant expressing human GDM-1 cell line treated with SFX-01 or vehicle. Graph bars represent mean expression levels ( ± SEM; $n = 4$ biological replicates) and P values calculated by unpaired t test. (C) Cell cycle analysis of GDM-1 cells cultured with SFX-01. Bars represent mean percentage ( ± SEM), and significance compared to controls was calculated by one-way ANOVA with Dunnett's test. $n = 4$ for vehicle or 250 μM SFX-01 treatment, $n = 3$ for 1, 5, 25, or 100 μM SFX-01 treatment, biological replicates. (D) Colony-forming ability of GDM-1 cells treated with SFX-01. Bars represent mean normalized values ( ± SEM), and because of the low sample size and not assuming a normal Gaussian distribution, significance was calculated using a Kruskal–Wallis test with an uncorrected Dunn's test for comparison to the control group. $n = 2$ for vehicle treatment, $n = 3$ for 0.1 μM SFX-01 treatment ($P = 0.3957$), $n = 3$ for 1 μM SFX-01 treatment ($P = 0.2167$), $n = 5$ for 10 μM SFX-01 ($P = 0.0245$) and $n = 3$ for 100 μM SFX-01 ($P = 0.0018$) treatment, biological replicates. (E) Cell cycle analysis and (F) colony-forming ability of control (CB) or JMML patient-derived HSCs cultured with or without SFX-01. Clonogenicity of HSC was normalized to control cells, and bars represent mean ( ± SEM) and P values calculated by two-way ANOVA with Sîdak post hoc test. For (E): CB: $n = 3$ for vehicle, 5, or 25 μM SFX-01 treatment, and $n = 2$ for 100 μM SFX-01 treatment. JMML: $n = 7$ for vehicle or 5 μM SFX-01 treatment, and $n = 8$ for 25 or 100 μM SFX-01 treatment, biological replicates. For (F): CB: $n = 14$ for vehicle treatment, $n = 4$ for 10 μM SFX-01 treatment, $n = 17$ for 50 μM SFX-01 treatment, and $n = 19$ for 100 μM SFX-01 treatment. JMML: $n = 6$ for vehicle or 10 μM SFX-01 treatment, $n = 11$ for 50 μM SFX-01 treatment, and $n = 12$ for 100 μM SFX-01 treatment, biological replicates. (G) A heatmap displaying the significantly altered Shp2 substrates in CD11b+ myeloid bone marrow cells isolated from WT or Ptpn11$^{D61G(-/+)}$ mice, under conditions with or without SFX-01 treatment based on data from the phosphoproteomics analysis ($n = 3$ biological replicates). Source data are available online for this figure.

The data presented here demonstrate that SFN and its derivative SFX-01 can inhibit Shp2 activity via covalent modification of cysteine residues within the catalytic domain of this phosphatase (Boivin et al, 2008; Hof et al, 1998; Weibrecht et al, 2007). This includes the formation of a dithiolethione bridge between Cys-333 and Cys-367, which lies proximal to the active site cysteine (Cys-459). While these two cysteines are not catalytic, their oxidation to an intramolecular disulfide inhibits phosphatase activity (Chen et al, 2009; Machado et al, 2017), and this is likely mimicked by the dithiolethione. The occurrence of this oxidative modification was verified using LC-MS/MS and thiol-specific labeling assays. This mode of inhibition is noteworthy because it circumvents the structural limitations imposed by the open conformation of mutant Shp2 variants, indicating SFN and SFX-01 possess the ability to inhibit even hyperactive mutants that are resistant to allosteric blockade.

The observed time-dependent loss of SFN immunoreactivity, in the context of persistent inhibition of phosphatase activity, is consistent with a model in which the initial SFN adduct transitions to a more stable dithiolethione product. This conversion alters the epitope recognized by the cysteinyl-SFN-specific antibody, and this conclusion was supported by studies in mutant HEK 293 cells expressing cysteine-to-serine Shp2 variants, which retained SFN labeling in the absence of one of the auxiliary cysteines required for dithiolethione formation. These findings provide mechanistic insight into how electrophilic inhibition by SFN may persist in vivo, even in the absence of detectable covalent adduction, and reinforce the potential of targeting catalytic cysteines as a therapeutic strategy.

Importantly, administration of SFX-01 to mice harboring the disease-causing D61G Shp2 mutation resulted in decreased activity of this phosphatase across multiple tissues, including the spleen and heart. This was accompanied by phenotypic improvements such as reduced leukocyte counts, decreased myeloid cell populations and normalization of spleen size. These therapeutic effects closely paralleled the observed biochemical inhibition of Shp2 and were not associated with overt toxicity, as assessed by comprehensive analyses of both plasma biochemistry and gross pathology. Furthermore, although chronic exposure to SFX-01 increased food intake and weight gain in WTs, it did not exacerbate any known

pathological features in NS model mice. This highlights a favorable tolerability profile of SFX-01 in genetic scenarios of increased Shp2 activity, consistent with studies in humans demonstrating safety and investigating therapeutic efficacy (Clack et al, 2025; Howell et al, 2019; Long et al, 2024; Zolnourian et al, 2024; Zolnourian et al, 2020). We speculate that the increased food consumption and growth in the WT may be connected with sulforaphane from SFX-01 altering their gut microbiome to influence food intake (Cardozo et al, 2021), although we note that the electrophile did not have any adverse effect on the small intestine of either genotype. It is notable that Noonan syndrome is associated with feeding difficulties, gut dysmotility and behavioural disorders causing avoidant or restrictive intake of some food types (Dumont et al, 2024; Shah et al, 1999). However, it remains unclear why SFX-01 increased food consumption and weight gain in WT and why this did not occur in the transgenics.

The exposure of mice in utero during development to SFX-01 yielded unexpected and adverse outcomes, including embryonic lethality in both homozygous and heterozygous D61G Shp2 offspring, particularly at higher drug concentrations. Mechanistic investigations indicated that SFX-01 exposure during gestation caused elevated ERK phosphorylation in neonates, which may have interfered with developmental signaling processes. Considering the extensive evidence that overactivation of ERK during development is maladaptive (Krenz et al, 2008; Krenz et al, 2005; Nakamura et al, 2007), this likely explains the decreased embryo viability caused by SFX-01. It is difficult to understand why administering SFX-01 to mothers carrying offspring from HET/HET crosses delivers no pups. This complete loss of offspring only occurred at the 2.5 mg/ml SFX-01 concentration, whereas some WT offspring were delivered when the lower 0.8 mg/ml concentration was used. It is possible that SFX-01-induced loss of transgenic mice caused stress that indirectly results in loss of WT mice that otherwise would be viable. Indeed, WT/WT crosses exposed to SFX-01 in utero do not result in a decrease in litter size, and so the losses relate to the presence of the mutant Ptpn11 gene. However, the situation is complex, as the loss in utero may be linked to the mother's genotype. Losses occur only when she carries the HET but not the WT genotype. This could be associated with the smaller size of the syndromic mice and their reduced ability to carry litters, especially

when additional stress is introduced by pERK activation through SFX-01. Overall, a probable explanation for the 2.5 mg/ml dose of SFX-01 causing embryonic lethality may be because of potentiated pERK, but as no pups were delivered, this could not be assessed.

At the cellular level, SFX-01 induced STAT1 phosphorylation, suppressed cyclin D1 expression and promoted PKR activation, collectively leading to growth arrest in leukocytes and Shp2-mutant leukemia cell lines. These observations were corroborated by phosphoproteomics analysis, which revealed decreased Shp2 phosphorylation at Y580, a site associated with phosphatase activation, and increased phosphorylation of known Shp2 substrates. This supports the conclusion that SFX-01-induced inhibition of Shp2 disrupts downstream growth signaling and contributes to the anti-proliferative phenotype observed in both in vitro and in vivo models. The phosphoproteomics studies also demonstrated Syk phosphorylation was modulated in the NS model $Ptpn11^{D61G(-/+)}$ mice when SFX-01 was administered. This is notable because this non-receptor tyrosine kinase, which is regulated by Shp2 redox state (Chen et al, 2025), controls platelet aggregation that is dysregulated in mice and humans with this syndrome (Artoni et al, 2014; Bellio et al, 2019). However, as Syk demonstrated complex phosphorylation alterations upon exposure to SFX-01, including at non-tyrosine residues, there is again complexity with defining the significance of these observations. Further complexity arises because SFX-01 modifies other cellular proteins, including signaling by the related phosphatase Shp1. The lack of genotype-specific differences in Shp1 Y564 phosphorylation upon SFX-01 treatment suggests a broader, though still biologically relevant, range of molecular targets. Nonetheless, the observed anti-proliferative effects in $Ptpn11^{D61G(-/+)}$ mice and JMML patient-derived hematopoietic stem cells support the notion that Shp2 remains a principal therapeutic target. Comparative experiments with SHP099 showed that combined treatment with both inhibitors did not produce additive effects, consistent with overlapping mechanisms of action and reinforcing the conclusion that the therapeutic effects of SFX-01 are primarily mediated through Shp2 inhibition.

To measure the selectivity of a drug requires knowledge of its multiple targets, as this enables comparative concentration-response studies. However, selectivity studies typically only assess related family members and mostly do not consider other protein classes are likely to be targeted and contribute to its biological actions. This is perhaps because a full list of molecular targets of a drug is elusive and difficult to determine. Most drugs are developed by screening an individual target, but thereafter, there is no routine method for unbiasedly identifying other proteins that it also binds to that may contribute to its cellular actions. In contrast, the natural compound SFN was not developed to a specific target, but because of its potential therapeutic value, it was rational to develop the stabilized SFX-01 derivative suitable for clinical use. As SFN covalently conjoins its targets, this allowed their identification as described herein, but this information, which would not normally be available for conventional drugs, raises the difficult question as to their potential role in the actions of the drug. Thus, the comprehensive identification of the targets of SFN has culminated in our understanding that other targets may contribute to the therapeutic actions of this electrophile. Although expression of a hyperactive Ras mutant theoretically provides a strategy for

defining whether a drug, here SHP099 or SFX-01, is dependent on Shp2 inhibition, the utility of this approach can be questioned, as discussed above.

Considering the selectivity of SFX-01 further, it is notable that it significantly decreased the phosphorylation of Shp2 Y580 in CD11b$^+$ myeloid bone marrow cells from NS model mice, but not those isolated from WTs. Although this may be because the D61G mutation increases the nucleophilicity of the Shp2 catalytic cysteine or access to it because of altered hydrogen bonding (Qiu et al, 2014), experimental evidence for this conjecture is lacking. Such alterations not only explain the enhanced phosphatase activity of the D61G mutant but are also anticipated to enhance its reactivity with the electrophile SFX-01 compared to WT Shp2. This would provide a mechanism that would make the electrophile more selective for the mutant phosphatase compared to WT. Whilst it is evident from the proteomics identifications that SFX-01 has multiple targets, the phosphatase activating mutation in the D61G mutant provides a rational basis for enhanced, selective inactivation by this electrophile compared to WT Shp2. Again, this is only speculation because we have no experimental evidence that D61E Shp2 is more sensitive to adductive inhibition by SFX-01 than the WT phosphatase. Shp1 also showed decreased phosphorylation of regulatory Y564 upon exposure to SFX-01, consistent with it also being targeted by this electrophile. Interestingly, and perhaps supporting SFX-01 being more selective for D61G than WT Shp2, the decrease in Shp1 Y564 phosphorylation induced by SFX-01 was comparable in each genotype. This provides additional support for the concept discussed that D61G Shp2 is more nucleophilic at the cysteine catalytic center than WT, explaining the enhanced phosphatase activity as well as selective electrophilic addition and inhibition by SFX-01. Thus, whilst SFX-01 can modify multiple targets, one of them is Shp2, and the D61G variant is potentially more susceptible.

Although SFX-01 interacts with multiple molecular targets, making it challenging to determine the exact contribution of Shp2 inhibition to its overall therapeutic effects, the evidence clearly indicates that inhibition of this phosphatase is a significant mechanism of action. As discussed, the broad reactivity of SFN and SFX-01 with cysteine-containing proteins introduces complexity in defining its full spectrum of biological effects. Whilst the covalent mechanism of action of SFN enabled the identification of its protein targets via chemical proteomics, this represents a double-edged sword. On one hand, it facilitates mechanistic clarity, but on the other, it challenges the attribution of specific phenotypic outcomes to individual protein modifications. Nevertheless, this approach highlights the potential of electrophilic compounds as targeted therapies in diseases driven by redox-sensitive signaling enzymes such as Shp2.

In conclusion, SFX-01 inhibits Shp2 through a novel covalent dithiolethione modification, enabling selective targeting of disease-associated mutant variants while exhibiting therapeutic efficacy in preclinical models of NS or in human JMML cells. The strong safety record of SFX-01 in adult mice, including NS models, together with solid evidence of its tolerability in humans (Clack et al, 2025; Howell et al, 2019; Long et al, 2024; Zolnourian et al, 2024; Zolnourian et al, 2020), and its ability to reverse Shp2-driven myeloproliferative disease, make it a promising candidate for clinical trials aimed at treating disorders caused by mutations that activate this phosphatase.

# Methods

## Reagents and tools table

| Reagent/resource | Reference or source | Identifier or catalog number |
| --- | --- | --- |
| **Experimental models** | | |
| B6;129S4-Ptpn11tm1Bgn/Mmjax | The Jackson Laboratory | 032104-JAX |
| HEK 293 | ATCC | CRL-1573 |
| GDM1 cell line | ATCC | CRL-2627 |
| **Recombinant DNA** | | |
| Shp2 WT | Addgene | #8381 |
| C459S Shp2 | Addgene | #8382 |
| **Antibodies** | | |
| anti-SFN primary antibody | In-house | N/A |
| Anti-ATP5A antibody | Abcam | ab14748 |
| Cy™3 AffiniPure® Goat Anti-Rabbit IgG (H + L) | Jackson Immuno Research Inc. | 111-165-144 |
| Cy™3 AffiniPure® Goat Anti-Mouse IgG (H + L) | Jackson Immuno Research Inc. | 115-165-003 |
| CD34 APC-eFlour780 | eBioscience | 47-0349-42 |
| CD8 | Biolegend | 301006 |
| CD20 | Biolegend | 302304 |
| CD2 | BD | 555326 |
| CD3 | BD | 345763 |
| CD16 | eBioscience | 11-0168-41 |
| CD19 | eBioscience | 11-0199-42 |
| CD235a | eBioscience | 11-9987-82 |
| CD66b | Biolegend | 305104 |
| CD10 | Biolegend | 312208 |
| CD127 | eBioscience | 11-1278-42 |
| Shp2 | R&D Systems | AF1894 |
| GAPDH | Cell signalling | CST #2118 |
| Anti-rabbit secondary antibody | Cell signalling | CST #7074 |
| CD11b MicroBeads | Miltenyibiotec. | 130-097-142 |
| Anti-Shp2 primary antibody | Santa Cruz | SC-7384-AC |
| CD11b-ApcCy7 | BD | 561039 |
| Ly6C- PerCP | BD | 561103 |
| Ly6G-FITC | BD | 561105 |
| Zombie Aqua | Biolegend | 423101 |
| Phosphorylated ERK 1/2 | Cell signalling | #9101 |
| Total ERK 1/2 | Cell signalling | #9102 |
| **Oligonucleotides and other sequence-based reagents** | | |
| Oligonucleotide primers | This study | Materials and methods |
| **Chemicals, enzymes, and other reagents** | | |
| 1,4 Dithiothreitol (DTT) | Roche | 10197777001 |
| D,L-Sulforaphane | Sigma-Aldrich | 574215 |
| Dimethyl sulfoxide | Sigma-Aldrich | D8418 |
| DMEM, high glucose, GlutaMAX™ Supplement, pyruvate | Thermo Fisher Scientific | 10569010 |
| Fetal bovine serum | Gibco | A5256801 |
| Pen Strep | Gibco | 15140-122 |
| R,S-Sulforaphane | LKT Labs | s8044 |
| S-Sulforaphane | LKT Labs | s8045 |
| R-Sulforaphane | LKT Labs | s8046 |
| SFX-01 | Theracryf plc | N/A |
| Maleimide | Sigma-Aldrich | 129585 |
| 4% paraformaldehyde | Sigma-Aldrich | P6148 |
| DPBS | Gibco | 14190094 |
| Bovine serum albumin | Fisher BioReagents | BP9702 |
| DAPI | Sigma-Aldrich | D9542 |
| DpnI | New England Biolabs | R0176S |
| 5-alpha Competent *E. coli* | New England Biolabs | C2987H |
| BD™ CompBeads | BD | 552843 |
| MethoCult™ H4435 Enriched | STEMCELL Technologies | N/A |
| RPMI 1640 | Gibco | 11835030 |
| Shp2 recombinant protein | Abcam | 32083 |
| TCEP | Thermo Fisher Scientific | T2556 |
| A/G plus agarose beads | Santa Cruz | #sc-2003 |
| SHP099 | Cayman Chemical | 20000 |
| WST-8 | Abcam | ab228554 |
| DiFMUP (6,8-Difluoro-4-Methylumbelliferyl Phosphate) | Thermo Fisher Scientific | D6567 |
| bisphosphorylated IRS1 peptide | BPS Bioscience | N/A |
| Biotin-iodoacetamide | Sigma-Aldrich | B2059 |
| HRP-conjugated streptavidin | Thermo Fisher Scientific | N100 |
| Biotinylated-phenylarsenic acid | SynInnova | N/A |
| Triethylammonium bicarbonate | Sigma-Aldrich | T7408 |
| Acetonitrile | Sigma-Aldrich | 34851 |
| Wright-Giemsa Stain | Sigma-Aldrich | WG80 |
| Red blood cell lysis buffer | Biolegend | 420301 |
| **Software** | | |
| Confocal microscopy | Leica SP5 system | N/A |
| FACS Aroa Fusion | BD | N/A |
| U3000 UHPLC NanoLC system | Thermo Fisher Scientific | N/A |
| VisualSonics Vevo 770 imagine system | FUJIFILM VisualSonics | N/A |
| GraphPad Prism 9 software | GraphPad | N/A |

| Reagent/resource | Reference or source | Identifier or catalog number |
|---|---|---|
| **Other** | | |
| Lyophilized Keyhole Limpet Hemocyanin | Calbiochem | H7017 |
| PD-10 desalting columns | Cytiva | 17085101 |
| NHS-activated sepharose | Cytiva | N/A |
| Q5 High-Fidelity PCR Kit | New England Biolabs | E0555S |
| Mycoplasma PCR Detection Kit | Abcam | ab289834 |
| QIAprep Spin Miniprep Kit | Qiagen | 27104 |
| Zeba spin columns | Thermo Fisher Scientific | 10415545 |
| 40-μm nylon mesh cell strainer | Thermo Fisher Scientific | 10737821 |

## Generation of anti-sulforaphane pan-specific antibody

Lyophilized Keyhole Limpet Hemocyanin (KLH, Calbiochem) was denatured and reduced (50 mM Tris pH 6.5, 150 mM NaCl, 100 mM dithiothreitol (DTT), 1% SDS) for 1 h at room temperature. The reduced, denatured KLH (20 mg. ~50 nmoles) was buffer exchanged into 50 mM Tris pH 6.5, 150 mM NaCl, 1% SDS pH 8.0 using PD columns (GE Healthcare) principally to remove DTT. The KLH was then reacted with a 3500-fold molar excess of DL-sulforaphane (Sigma) dissolved in DMSO. The mixture remained clear and free of precipitate, and the reaction was allowed to proceed overnight at room temperature with gentle agitation. Aliquots of the mixture were spin dried by rotary evaporation (Eppendorf), the pellets washed with water to remove excess DMSO, and re-dried by freeze drying (Edwards Modulyo) to generate the antigen. Four rabbits were immunized with a total of 3 mg of the antigen emulsified in complete Freund's adjuvant at 5 time points over 77 days (Cambridge Research Biochemicals). Development of the antibody response was monitored by ELISA screening, and terminal bleeds provided $4 \times 50$ ml antisera. The anti-SFN primary antibody was affinity-purified from rabbit serum, which had been stored at $-80\,°C$ since harvesting. An affinity column using N-hydroxysuccinimide NHS-activated Sepharose (GE Healthcare) was made via the batch method. In total, 5 ml of activated Sepharose was washed in $15 \times 10$ ml cold 1 mM HCl and incubated with 50 mM L-cysteine in 8 ml coupling solution (0.2 M NaHCO$_3$, 0.5 M NaCl, pH 8.3). The matrix mixture was rotated overnight at $4\,°C$, and the following morning centrifuged at $1000 \times g$ for 1 min, and the supernatant was discarded. Beads were resuspended in 8 ml 0.1 M Tris-HCl pH 7.5 and incubated for 2 h at room temperature. Rapid alternate washes were carried out using 0.1 M Tris-HCl pH 8.5 followed by 0.1 M acetate buffer pH 4.0 plus 0.5 M NaCl, repeated 3X and washed in $5\times10$ ml 10 mM DTT in PBS pH 8.0. 5 ml 50 mM L-SFN was added and incubated overnight at $4\,°C$. The following morning, the column was washed with $20 \times 10$ ml PBS supplemented with 20% DMSO, then $5\times 10$ ml PBS. In all, 10 ml of rabbit antiserum was added and rotated overnight at $4\,°C$. The following morning, beads were poured into a PD-10 column (GE Healthcare) and the antiserum was passed through the

column. The column was washed 3× with 10 ml PBS plus 500 mM NaCl, followed by 10 ml PBS. 17 ml of acid-elution buffer (100 mM glycine, pH 2.5) was added to the column and collected in a 15-ml falcon tube containing 3 ml 1 M Tris, pH 8.0, neutralized with 9 ml PBS followed by 17 ml of basic-elution buffer (100 mM triethylamine, pH 11.5) collected in a 15 ml falcon tube containing 3 ml 1 M acetate pH 5.0. The column was washed with 9 ml PBS and stored in PBS plus 0.05% azide at $4\,°C$. The two pools of antibody were then concentrated using spin columns with a 50 K cut-off (Millipore) via centrifugation at $3000 \times g$ at $4\,°C$ into storage buffer (PBS, 0.1% Tween-20, 1% trehalose, 0.01% azide). Pools were concentrated until optical density at 280 nm exceeded 1 (where 1 OD at 280 nm = 0.7 mg/ml IgG). The purified antibody was stored at $-20\,°C$ until required for immunolabelling or immunoprecipitation protocols. All subsequent studies were only performed with IgG eluted under basic conditions as this pool demonstrated the highest immunoreactivity by immunoblotting. The anti-sulforaphane antibody is available on request.

## Culture of HEK 293 cells

Cells were sourced from the American Type Culture Collection and greatly expanded in culture to provide substantive numbers of aliquoted, frozen stocks that are regularly accessed to ensure that passage number is low, and cell identity is maintained. Cell authenticity was maintained by recurrent purchase from the supplier and was routinely tested for mycoplasma using the Mycoplasma PCR detection kit (Abcam). HEK 293 cells were maintained in DMEM plus GlutaMAX-I (Thermo Fisher Scientific), supplemented with 10% fetal bovine serum (FBS) and penicillin/streptomycin (1 unit/ml;1 μg/ml) in a 95% O$_2$/5% CO$_2$ incubator at $37\,°C$. Cells were maintained in a T75 flask and washed, detached using trypsin and seeded in a six-well plate when required for experiments. In some experiments, cells were treated with 1–10 μM of racemic sulforaphane (RS-) or the individual (R-, S-) isomers (LKT Laboratories) for 20 min, or 1–250 μM SFX-01 for 0.5–4 h followed by lysis via sonication for 7 s at 30 kHz and 40% amplitude in 2× SDS-PAGE sample buffer (50 mM Tris-HCl pH 6.8, 2% SDS, 0.1% bromophenol blue, 10% glycerol) c/w 100 mM maleimide.

## Immunostaining

HEK 293 cells were fixed with 4% paraformaldehyde (Sigma) in PBS for 10 min at room temperature, washed in PBS and permeabilised with 0.2% Triton X-100 in PBS for 5 min. In order to avoid non-specific antibody binding the cells were incubated with blocking buffer (5% non-specific goat serum diluted in 1% bovine serum albumin, 20 mM Tris pH 7.5, 155 mM NaCl, 2 mM EGTA, 2 mM MgCl$_2$) for 1 h at room temperature. Cells were incubated in primary antibody [Sulforaphane IgG (1:100) or ATP5α IgG (1:100, Abcam)] diluted in blocking buffer overnight at $4\,°C$ and washed with PBS the next day. For double immunofluorescence, conjugated secondary antibodies (1:100, Jackson ImmunoResearch) were used for 1 h at room temperature [Cy3 anti-rabbit (111-165-144) for sulforaphane IgG, and Cy5 anti-mouse (115-177-003) for ATP5α IgG]. Diamidine-2-phenylindole dihydrochloride (DAPI) (Sigma) was added with the secondary antibodies to stain the nucleus. The specimens were analyzed using

confocal microscopy on an inverted microscope (Leica SP5 system) equipped with a blue diode and argon and helium neon lasers using a ×63/1.4 numerical aperture oil immersion lens.

## Site-directed mutagenesis

WT or C459S Shp2 plasmids were purchased from Addgene (plasmid #8381 and #8382, respectively). C333/367S and C333/459S were generated by site-directed mutagenesis using the oligonucleotide primers, and PCR was carried out using a Q5 High-Fidelity DNA Polymerase kit (New England Biolabs). For C333S mutant: forward primer for PCR 5′- ACA-CAAGGCTCCCTGCAAAAC-3′ reverse primer for PCR 5′-GGCAATGTAACTCTTTTTGG-3′. For C333/367S mutant: forward primer for PCR 5′-AAGAGTAAATCTGTCAAATACTGGC-3′ reverse primer for PCR 5′- TCCTCTCTCCACTTCTTTC-3′. For C333/459S mutant: forward primer for PCR 5′ -ACA-CAAGGCTCCCTGCAAAAC-3′ reverse primer for PCR 5′-GGCAATGTAACT CTTTTTGG-3′. Following PCR DpnI (New England Biolabs) was added to the mixture and incubated at 37 °C for 1 h. In all, 5 µl of PCR mix was added to one vial of 5-α competent *Escherichia coli* cells (New England Biolabs), incubated on ice for 30 min, heat shocked at 42 °C for 30 s followed by a final incubation on ice for 5 min. In total, 500 µl of SOC medium was added to the competent cells, incubated at 37 °C for 1 h shaking at 300 rpm and 100 µl were spread onto an LB agar plate supplemented with 100 mg/ml ampicillin and incubated at 37 °C overnight. The following morning 5 ml of LB broth supplemented with 100 mg/ml ampicillin was inoculated with single bacterial colonies incubated at 37 °C overnight shaking at 100 rpm. The following morning, the plasmid was extracted from the bacterial cells using a plasmid miniprep kit (QIAGEN) and the DNA was sequenced (Eurofins Genomics). The mutant Shp2 plasmids are available on request.

## Patient sample collection

Patient samples were collected (Patient ID5: Mutation PTPN11 c.1508 G > C; Patient ID6: Mutation PTPN11 c.227 A > C; Patient ID22: Mutation PTPN11 c.226 G > A; Patient ID7: Mutation PTPN11 c.227 A > C, SRSF2 c.287 C > T; Patient ID23: Mutation KRAS c.179 G > A, PTPN11 c.215 C > T), following informed consent, in accordance with the principles set out in the WMA Declaration of Helsinki and the Department of Health and Human Services Belmont Report for sample collection and use in research, under the UK NIHR NHS REC approved Paediatric MDS/JMML study (NHS REC reference number 15/LO/0961. Cord blood samples were commercially sourced (Zenbio US, Cat# SER-CD34-F). Patient's bone marrow samples were screened for JMML mutations using a customized JMML next-generation sequencing panel, as previously described (Louka et al, 2021).

## Fluorescence-activated cell sorting

FACS-sorting of mononuclear cells was performed using FACS Aria Fusion II (Becton Dickinson). All experiments included single color-stained controls (CompBeads, BD Biosciences) and Fluorescence Minus One controls (FMO). JMML lin-CD34+ cells were isolated from JMML patient samples and normal controls, using

antibodies CD34 APC-eFlour780 (1:200, eBioscience, Clone: 4H11, Cat#: 47-0349-42, RRID: AB_2573956), CD8 (1:200, Biolegend, Clone: RPA-T8, Cat#: 301006, RRID: AB_314124), CD20 (1:200, Biolegend, Clone: 2H7, Cat#: 302304, RRID: AB_314252), CD2 (1:200, BD, Clone: RPA-2.10, Cat#: 555326), CD3 (1:100, BD, Clone: SK7, Cat#: 345763), CD16 (1:200, eBioscience, Clone: eBioCB16, Cat#: 11-0168-41, RRID: AB_10804882), CD19 (1:200, eBioscience, Clone: HIB19, Cat#: 11-0199-42, RRID: AB_10669461), CD235a (1:200, eBioscience, Clone: HIR2, Cat#: 11-9987-82, RRID: AB_465477), CD66b (1:200, Biolegend, Clone: G10F5, Cat#: 305104, RRID: AB_314496), CD10 (1:200, Biolegend, Clone: HI10a, Cat#: 312208, RRID: AB_314919), CD127 (1:200, eBioscience, Clone: RDR5, Cat#: 11-1278-42, RRID: AB_1907343), each FITC.

## Clonogenic assays with primary JMML cells

Lin-CD34+ cells were sorted from either JMML patient samples or cord blood controls into 96-well plates with 50 µL of Methocult H4435 (Stemcell Technologies), at a concentration of 10 cells/well and were cultured at 37 °C and 5% CO$_2$ for 14 days. SFX-01 was added in different concentrations to methocult prior to sorting. Clonogenic capacity was assessed on day 14 by evaluating the morphology and presence of colonies under direct light microscopy.

## Culture of GDM1 cells

GDM1 cell line was purchased ATCC (CAT CRL-2627) and maintained in RPMI (Gibco) + 20% FBS + 1% penicillin–streptomycin (Sigma-Aldrich). GDM1 cells were plated in Methocult H4435 (Stemcell Technologies) at a concentration of 200 cells per well and were cultured at 37 °C and 5% CO$_2$. Clonogenic capacity was assessed after 14 days of cell culture and individual colonies were picked for cytospins.

## Immunoblotting

Protein-containing samples in SDS-PAGE sample buffer were subjected to electrophoresis and immunoblotting using the following primary antibodies: SFN (1:1000, in-house), Shp2 (1:1000, Abcam #32083) for recombinant protein or (R&D Systems #AF1894) for immunoprecipitation experiments, GAPDH (1:5000, CST #2118) and an anti-rabbit secondary antibody (1:2500, CST #7074).

## Animals

All procedures were approved by the Queen Mary University of London and the King's College London animal welfare and ethical review body and were performed in accordance with the Home Office Guidance on the Operation of the Animals (Scientific Procedures) Act 1986 in the United Kingdom under Project Licence PB137135C. *Ptpn11D61G(−/+)* mice (Strain Name: B6;129S4-Ptpn11tm1Bgn/Mmjax, Stock Number: 032104-JAX) were purchased from the Mutant Mouse Regional Research Centre. In all, 12-week-old male and female mice were used for experiments. Mice were maintained in a Biological Services Unit under specified pathogen-free conditions with ad libitum access to

chow and water and a 12-h day/night cycle at 20–22 °C at 60% humidity.

## Identification of proteins adducted by SFN after its oral administration to mice

Hearts, isolated from mice 3 h after they were orally gavaged with 8 mg SFN, were frozen in liquid nitrogen, under which they were crushed to a powder using a pestle and mortar. This tissue was then resuspended and gently agitated in RIPA buffer (50 mM Tris pH 7.5, 150 mM NaCl, 1% Triton X-100, 0.5% sodium deoxycholate, 0.1% SDS) containing 5 mM EDTA, 5 mM EGTA, 100 mM maleimide, 25 mM TCEP, and protease inhibitors. Extracts were clarified by centrifugation for 5 min at $25,000 \times g$ before buffer exchange into immunoprecipitation buffer (RIPA with 10 mM maleimide and protease inhibitors) using Zeba spin columns (Thermo) to remove TCEP and excess maleimide. The anti-SFN antibody (1:100) was incubated with SFN-adducted proteins overnight at 4 °C. The antibody conjugates were isolated by incubation with protein A/G beads (20 µl per immunoprecipitation reaction, Santa Cruz) for 2 h, which were washed 4× with immunoprecipitation buffer before proteins were eluted from the beads by adding 120 µl non-reducing SDS sample buffer containing 100 mM maleimide and stored frozen until further analysis.

Cardiac immunoprecipitates from four vehicle-control or four SFN-treated hearts were separated on a single 10% SDS-PAGE gel prepared the day before to minimize contamination with unreacted acrylamide. For mass spectrometry, each entire lane was sliced into seven consecutive pieces. Gel slices were reduced, alkylated and digested with trypsin and modified for use with an Investigator ProGest (Genomic Solutions) robotic digestion system. Digested peptides were separated by nanoflow liquid chromatography (Ultimate 3000™, Dionex) on a reverse-phase column (PepMap 100, 75 µm I.D., 25 cm length, Dionex) at a flow rate of 300 nl/min. The column was coupled to an Orbitrap mass spectrometer (LTQ Orbitrap XL, Thermo Fisher Scientific), utilizing full ion-scan mode over the mass-to-charge ($m/z$) range 400–1600. Tandem mass spectrometry (MS/MS) was carried out on the top six ions in each MS scan using the data-dependent acquisition mode with dynamic exclusion enabled. Generated spectra were matched to database entries (UniProt/SwissProt mouse database 2014_01, 16656 protein entries) using Mascot 2.3.01 (Matrix Science). Scaffold (version 4.3.2, Proteome Software Inc., Portland, OR) was used to validate MS/MS-based peptide and protein identifications. Protein identifications were accepted if peptide probability was greater than 90%, protein probability was greater than 99.9% and with a minimum of two peptide spectrum matches (PSMs). For any one protein, peptides were derived from up to four control or four SFN-treated hearts, and the PSMs from the eight separate hearts were indexed separately. This allowed for a statistical analysis with a two-tailed $t$ test between the four control or treated groups. For each identified protein, the numbers of PSMs from the four control and treated hearts were summed, respectively, and enrichment within the treated compared to the control group was estimated (#PSMs in treated hearts/#PSMs in control hearts), with the total adjusted to 0.1 where no PSM was identified. The data was plotted as a volcano plot with relative enrichment plotted on the $x$ axis in the form of binary log of (# treated PSMs/# control PSMs), and $P$ value confidence plotted on the $y$ axis in the form of (−) natural log $P$

value. Protein hits were considered significant if the enrichment was at least fourfold ($x \geq 2$) and the confidence was over 95% ($y \geq 1.3$).

## CD11b+ bone marrow cell isolation and cell proliferation assay

Bone marrow mononuclear cells were isolated from femurs and tibia of WT or $Ptpn11^{D61G(-/+)}$ mice. Cluster of differentiation molecule 11b (CD11b) + cells were subsequently isolated using CD11b microbeads (1:10, Miltenyi Biotec) according to the manufacturer's instructions. The purified CD11b+ myeloid bone marrow cells were treated with PBS or 10 µM SFX-01 for 2 h in serum-free media. Following treatment, cells were lysed with 8 M urea buffer containing sodium orthovanadate, sodium fluoride, β-glycerol phosphate, and disodium pyrophosphate and sonicated for phosphoproteomics analysis. For the cell proliferation assay, CD11b+ bone marrow cells were seeded at $5 \times 10^4$ cells per well in a 96-well plate. The following day, cells were treated with 1 µM or 10 µM SFX-01, 10 µM SHP099 or a combination of these compounds, and incubated for 24 h. After incubation, WST-8 reagent (Abcam) was added to each well and the measured at 450 nm (BMG Labtech) to assess cell proliferation.

### Phosphoproteomics experiment

Phosphoproteomics experiments were performed using mass spectrometry with some technical modifications as reported (Casado et al, 2013; Hijazi et al, 2020). In brief, frozen cell pellets were lysed in 8 M urea buffer and supplemented with phosphatase inhibitors (10 mM $Na_3VO_4$, 100 mM β-glycerol phosphate and 25 mM $Na_2H_2P_2O_7$ (Sigma)). 110 µg of proteins were digested into peptides using trypsin as previously described (Alcolea et al, 2012; Montoya et al, 2011). Phosphopeptides were desalted and enriched using the AssayMAP Bravo (Agilent Technologies) platform. For desalting, the protocol peptide clean-up v3.0 was used. Reverse-phase S cartridges (Agilent, 5 µl bed volume) were primed with 250 µl 99.9% acetonitrile (ACN) with 0.1%TFA and equilibrated with 250 µl 0.1% TFA at a flow rate of 10 µl/min. The samples were loaded at 20 µl/min, followed by an internal cartridge wash with 0.1% TFA at a flow rate of 10 µl/min. Peptides were then eluted with 55 µl of solution (70/30 ACN/ $H_2O$ + 0.1% TFA) into a pre-filled well plate with 1 M glycolic acid with 50% ACN, 5% TFA. Following the Phospho Enrichment v 2.1 protocol, phosphopeptides were enriched using 5 µl Assay MAP TiO2 cartridges on the Assay MAP Bravo platform. The cartridges were primed with 100 µl of 5% ammonia solution with 15% ACN at a flow rate of 300 µl/min and equilibrated with 50 µl loading buffer (1 M glycolic acid with 80% ACN, 5% TFA) at 10 µl/min. Samples eluted from the desalting were loaded onto the cartridge at 3 µl/min. The cartridges were washed with 50 µl loading buffer and the phosphorylated peptides were eluted with 25 µl 5% ammonia solution with 15% CAN directly into 25 µl 10% formic acid. Phosphopeptides were lyophilized in a vacuum concentrator and stored at −80 °C. Dried phosphopeptides were dissolved in 0.1% TFA and analyzed by a nanoflow ultimate 3000 RSL nano instrument was coupled online to a Q Exactive plus mass spectrometer (Thermo Fisher Scientific). Gradient elution was from 3% to 28% solvent B in 60 min at a flow rate 250 nl/min with

solvent A being used to balance the mobile phase (buffer A was 0.1% formic acid in water and B was 0.1% formic acid in acetonitrile). The spray voltage was 1.95 kV, and the capillary temperature was set to 255 °C. The Q-Exactive Plus was operated in data-dependent mode with one survey MS scan followed by 15 MS/MS scans. The full scans were acquired in the mass analyser at 375–1500 $m/z$ with the resolution of 70,000, and the MS/MS scans were obtained with a resolution of 17,500. For data analysis for phospho and total proteomics, MS raw files were converted into mzML using MSConvert, part of the ProteoWizard software package (Holman et al, 2014). FragPipe (version v22.0) (Yu et al, 2021) with MSFragger (version 4.1) (Kong et al, 2017), Percolator (3.07.01) (Käll et al, 2007), and PTM-Shepherd (v2.0.5) (Geiszler et al, 2021) to analyze the data. Peptide and phosphopeptide quantification were performed using in-house software Pescal as described before (Alcolea et al, 2012). The resulting quantitative data was further analyzed using Protools 2R package (https://github.com/CutillasLab/protools2/releases/tag/v0.2.12).

## Treatment of WT or *Ptpn11*$^{D61G(-/+)}$ mice with SFX-01

WT or *Ptpn11*$^{D61G(-/+)}$ mice received 2.5 mg/ml SFX-01 (~40 mg/kg/day SFN) in their drinking water for 4–10 days or 10 weeks. SFN is stable in water for at least 6 days (Appendix Fig. S5), and the water containing the SFX-01 was replaced every 4 days. At the end of in vivo experiments, mice were sacrificed using a single intraperitoneal injection comprised of 70% sodium pentobarbitone and 30% sodium heparin. Tissue was isolated, flushed with saline, frozen by immersion in liquid nitrogen and homogenized in 10% wt/vol ice-cold homogenization buffer (100 mM Tris-HCl, 150 mM NaCl, 1 mM EGTA, 1% Triton, pH 7.4) supplemented with metal-chelator-free protease inhibitor (Roche). For toxicity testing, the mouse body weight was measured every two weeks. Food consumption and water intake were recorded weekly or at the time of SFX-01 replacement for 10 weeks. Plasma samples were sent to the Medical Research Council (MRC) Harwell for comprehensive biochemical profiling. The analyses included the measurement of sodium, potassium, chloride, total calcium, inorganic phosphorus, iron, magnesium, creatinine (enzymatic), urea, uric acid, total protein, albumin, total bilirubin, alkaline phosphatase (ALP), alanine aminotransferase (ALT), aspartate aminotransferase (AST), lactate dehydrogenase (LDH), creatine kinase (CK), α-amylase, total cholesterol and glucose. These measurements were conducted using the Beckman Coulter DxC 700 AU Analyzer. The small intestine was removed and fixed in 4% paraformaldehyde. Tissue samples were afterwards embedded, sectioned, and stained with Hematoxylin and Eosin (H&E). All stained slides were scanned and analyzed using a NanoZoomer S210 (Hamamatsu). For fetal treatment with SFN, three genotype breeding pairings were used for foetal studies: WT/WT, WT/HET, with the male mouse being HET, and HET/HET. In preliminary studies, male and females were housed separately and received 2.5 mg/ml or 0.8 mg/ml SFX-01, which is equivalent to 0.385 mg/ml or 0.123 mg/ml SFN, respectively, for 3 days in their drinking water prior to breeding. Males were then added to the females' cages, where they remained until pregnancy was confirmed by visual inspection of the female. Dams continued receiving SFX-01 throughout their pregnancy. Neonates were sacrificed and snap frozen by liquid nitrogen within 24 h of birth. In additional studies, male and females were mated, and the date of conception

was calculated by identification of a vaginal plug and at this point, the males were removed. Treatment with 2.5 mg/ml SFX-01 in the dams drinking water began 11 days post-conception, the time point determined as the completion of gastrulation. Neonates were sacrificed and snap frozen by liquid nitrogen within 24 h of birth.

## Determining the stability of SFN in water

In all, 2.5 mg/ml SFX-01 (0.385 mg/ml SFN) dissolved in deionised water was stored in the dark at room temperature, and each morning (0–6 days) the solutions were analyzed by HPLC using the protocol below. The HPLC system (LC-10 AD liquid chromatograph, Shimadzu) consisted of a gradient pump used at a flow rate of 1 ml/min, an injection valve programmed to inject 20 µl of sample, a UV detector set to 205 nm and a Supelco reverse-phase Titan C-18 column (Sigma-Aldrich) held in a column oven set to 37 °C. The mobile phase consisted of buffer A (100% deionised water) and buffer B (90% acetonitrile:10% deionised water) following a binary gradient program.

## Shp2 inhibition assay

Shp2 catalytic activity was monitored using the surrogate substrate, 6,8-difluoro-4-methylumbelliferyl phosphate (DiFMUP) (Invitrogen). Reactions were performed in a 96-well, flat-bottom plate using a final reaction volume of 50 µl. In all, 25 µl of assay buffer (25 mM Tris-HCl, 75 mM NaCl, 2 mM EDTA, 0.16 mM DTT pH 7.2) was added to 25 µl of Shp2-containing solution of either: 0.011 ng/µl recombinant human Shp2 protein (Abcam) (desalted into an experimental buffer (200 µM DiFMUP, 25 mM Tris-HCl, 75 mM NaCl, 0.05% Tween-20, 2 mM EDTA, 0.16 mM DTT, pH 7.2)) which had been treated for 0.5, −6 h −/+ 0.007–1.75 µM SFX-01 and/or 0.5 µM of bisphosphorylated IRS1 peptide (sequence: H2N-LN(pY)IDLDLV(dPEG8)LST(pY)-ASINFQK-amide (BPS Bioscience), or Shp2 immunoprecipitated from cardiac or spleen tissue of WT or *Ptpn11*$^{D61G(-/+)}$ mice treated or untreated with SFX-01 for 4 days to 10 weeks, +/− subsequent 30 min treatment with 5 mM DTT. The fluorescence signal was monitored using a microplate reader (Gemini XPS) at an excitation of 360 nm and an emission of 460 nm at room temperature for 15 min. Overall, 0–100 µM DiFMU standards were used in each experiment and a line of best fit was generated. The amount of product formed from experimental samples was calculated by extrapolating the corresponding Y value from the line of best fit. DiFMUP or SFN alone were used as negative controls for in vitro experiments. Tissue lysate incubated with unconjugated agarose bead and antibody/bead were used as negative controls for in vivo studies.

## Immunoprecipitation of Shp2 protein

Shp2 was immunoprecipitated from cardiac tissue lysate of WT or *Ptpn11*$^{D61G(-/+)}$ mice treated or untreated with SFX-01 using agarose bead-conjugated anti-Shp2 primary antibody (Santa Cruz SC-7384-AC) rotating at 20 rotations per minute for 3 h at 4 °C. Subsequent wash steps were carried out using RIPA buffer (50 mM Tris pH 7.5, 150 mM NaCl, 1% Triton X-100, 0.5% sodium deoxycholate, 0.1% SDS). If immunoprecipitated protein was required for western blotting, samples were added to SDS-PAGE sample buffer, heated at 95 °C for 5 min to unconjugate the protein from the bead and

centrifuged at $15,000 \times g$ for 5 min. To analyze Shp2 phosphatase activity the protein was left conjugated to the bead.

## Biotin-iodoacetamide labeling

In all, 400 μM biotin-iodoacetamide was added to Shp2 immunoprecipitated from cardiac tissue of WT or $Ptpn11^{D61G(-/+)}$ mice treated or untreated with SFX-01 for 4 days and incubated at room temperature for 30 min followed by the addition of an equal volume SDS-PAGE sample buffer containing 100 mM maleimide. Samples were analyzed under non-reducing conditions by SDS-PAGE and western blotting. PVDF membranes were probed using HRP-conjugated streptavidin (1:10,000 in 5% BSA in PBS-T) for 1 h.

## Phenylarsenic acid labelling

Shp2 immunoprecipitated from cardiac tissue of WT or $Ptpn11^{D61G(-/+)}$ mice treated or untreated with SFX-01 for 4 days was added to an equal volume of SDS-PAGE sample buffer and analyzed under non-reducing conditions by SDS-PAGE and far-western blotting. PVDF membranes were incubated with 1 mM biotinylated-phenylarsenic acid (Synlnnova) in 5% BSA in PBS-T for 1 h at room temperature, washed with PBS-T and incubated for another hour with HRP-conjugated streptavidin (1:10,000 in 5% BSA in PBS-T).

## Identification of a dithiolethione following treatment of Shp2 with SFN in vitro

Overall, 15 μg of recombinant Shp2 protein (Sigma) treated with equimolar amounts of SFN for 6 h was electrophoresed into an SDS-PAGE gel and stained with a colloidal Coomassie stain. The Shp2 band was excised and destained with 100 mM triethylammonium bicarbonate (TEAB) and acetonitrile (50%:50%) in a shaking heat block at 37 °C until blue color was completely removed. The gel pieces were dehydrated using acetonitrile and dried in a SpeedVac. Trypsin was added at a 1:5 (enzyme:substrate) ratio and incubated at 37 °C overnight. The original supernatant from the trypsinised gel pieces was removed and transferred to a new 1.5 ml centrifuge tube. Peptides were extracted from the gel pieces using a series of acetonitrile dehydration and 50 mM TEAB hydration steps. The supernatant from each step was removed and pooled with the original supernatant. The total volume was dried to completion in a SpeedVac. The sample was resuspended in 30 μl sample loading buffer (2% acetonitrile/ 0.05% FA/Water). 15 μl was loaded onto the column for mass spectrometry analysis.

Chromatographic separation was performed using a U3000 UHPLC NanoLC system (Thermo Fisher Scientific). Peptides were resolved by reversed-phase chromatography on a 75 μm C18 column (50 cm length) using a three-step linear gradient of 80% acetonitrile in 0.1% formic acid. The gradient was delivered to elute the peptides at a flow rate of 250 nl/minute over 60 min. The eluate was ionized by electrospray ionization using an Orbitrap Fusion Lumos (Thermo Fisher Scientific) operating under Xcalibur v4.1.5. The instrument was programmed to acquire in automated data-dependent switching mode, selecting precursor ions based on their intensity for sequencing by collision-induced fragmentation using a TopN CID method. The tandem mass spectrometry analyses were conducted using collision energy profiles that were chosen based on the mass-to-charge ratio (m/z) and the charge state of the peptide.

A database was created from the sequence of Shp2 protein and uploaded into Proteome Discoverer (PD) version 1.4 and version 2.2. The raw data files were processed and searched using the Mascot search algorithm (v2.6.0; www.matrixscience.com) and the Sequest search algorithm against the bespoke Shp2 database. Modifications allowed in the search were oxidation on methionine and oxidation, deoxidation and trioxidation on cysteine. The raw data were processed through Proteome Discoverer v 2.3 with the inclusion of a node for detection of crosslinked peptides. This was performed to detect modified peptides that are bound between cysteine residues within the sequence. In silico digestion was performed to determine possible peptides containing cysteine residues after digestion with trypsin. The mass of these peptides, combined with the mass of the dithiolethione modification, was included. The theoretical masses were then searched with the XLinkX node in PD v2.3 utilizing the non-cleavable linker option to determine identification of possible matches to the mass of the parent ion in the mass spectrometry survey scan (MS1) and subsequent tandem mass spectrometry (MS2) fragmentation spectra. These peptides would not have ordinarily been assigned and matched in the conventional database search.

## Total white blood cell count

Total white blood cell count was analyzed by Wright-Giemsa staining of 1 μl of whole blood sampled from mice via the tail vein. White blood cells were counted from five fields of view at a ×10 magnification for all samples, and the white blood cell count was represented as an average of these five values.

## Tissue preparation for flow cytometry

Single-cell suspensions from blood, spleen and bone marrow were prepared as follows. All centrifugation steps were carried out for 10 min at $1800 \times g$ at 4 °C, and all incubation steps were carried out at room temperature.

## Blood: erythrocytes were lysed by using PBS and centrifuged

### Spleen
Spleen tissue was dissected into two halves along the long axis; half snap frozen at −80 °C whilst the remaining was passed through a 40-μm nylon mesh cell strainer (Thermo Fisher Scientific) into 25 ml PBS and centrifuged. Erythrocytes were lysed using red blood cell lysis buffer (BioLegend) and centrifuged.

### Bone marrow
Bone marrow was flushed from the femur and tibia bones into 25 ml PBS and centrifuged. Erythrocytes were lysed using red blood cell lysis buffer and centrifuged.

Pellets were resuspended in PBS, passed through a cell strainer into round-bottomed flow cytometry tubes and washed with flow cytometry buffer (2% FBS, 2 mM EDTA in PBS). Cells were subsequently resuspended in 200 μl flow cytometry buffer and incubated for 1 h in the dark with 1 μl of each antibody; CD11b-allophycocyanin Cy7 (ApcCy7, 1:200 dilution), lymphocyte antigen

### The paper explained

#### Problem

Activating mutations in Src homology-2 domain-containing protein tyrosine phosphatase-2 (Shp2) drive multiple conditions with limited treatment options. These include juvenile myelomonocytic leukemia (JMML) and Noonan syndrome, which can be caused by mutations in Shp2 that increase its phosphatase activity by enhancing the nucleophilic reactivity of its catalytic cysteine. SFX-01, an α-cyclodextrin-stabilized sulforaphane complex in clinical development covalently modifies cysteine residues, but its molecular targets are not fully elucidated

#### Results

To identify the molecular targets of sulforaphane, we developed an antibody tool that detects proteins that it covalently adducts to. This antibody also enabled immuno-affinity enrichment of modified proteins, which were then identified using liquid chromatography-tandem mass spectrometry (LC-MS/MS). Among the targets was the cysteine thiol-dependent protein phosphatase Shp2. SFX-01 induced an inhibitory dithiolethione modification at its active site cysteine. In a transgenic mouse model of Noonan syndrome with hyperactive D61G Shp2, SFX-01 restored normal phosphatase activity and reduced myeloid cell expansion. In JMML patient-derived hematopoietic stem cells, SFX-01 suppressed proliferation by inhibiting STAT1 signaling and reducing cyclin D1 expression, leading to cell-cycle arrest.

#### Impact

These findings establish SFX-01 as an inhibitor of activating mutants of Shp2 that may offer therapeutic benefits for patients with JMML or Noonan syndrome.

6 C (Ly6C)-peridinin-Chlorophyll-Protein (PerCP, 1:200), lymphocyte antigen 6C (Ly6G)-fluorescein isothiocyanate (1:200). All purchased from BD Biosciences. In total, 1 µl of Zombie Aqua (BioLegend) was also added to facilitate the exclusion of dead cells. Data were acquired using a FACSCanto II cell analyzer system (BD Biosciences) using a 633 nm excitation red laser to detect CD11b-ApcCy7, a 488 nm excitation red laser to detect Ly6C-PerCP and a 488 nm excitation green laser to detect Ly6G-fluorescein isothiocyanate.

## Ultrasound

Ultrasound was performed using the VisualSonics Vevo 770 imagine system fitted with an RMV707B scan head at 15-45 MHz. Mice anaesthetized using 3% isoflurane mixed with 97% $O_2$ at a flow rate or 1 l/minute and placed in a supine position on top of a heated pad and fixed in place. Anesthesia was maintained using a nose mask with 1.5–2% isoflurane and 98–98.5% $O_2$ at a flow rate of 1 l/min. Hair was removed from the abdominal area, and preheated ultrasound gel was applied. The probe was moved into position until the largest area of the long axis of the spleen was identified, which was measured in mm².

## Statistical analyses

Data are shown as mean ± SEM unless otherwise indicated in figure legends. The number of biological replicates for cellular experiments is expressed as $n$ in the figure legends. Statistical analyses

were conducted using GraphPad Prism 9 software. Recombinant Shp2 activity data (Fig. 2A,B) were fit to a one-phase exponential decay curve and calculated goodness of fit. Significance was assessed using paired $t$ tests for before-and-after plots and by two-way ANOVA with Sìdak multiple test correction. Statistical significance was determined by $P < 0.05$. No tests were conducted to assess inclusion or exclusion of data, and consistent with this, all observations were retained during statistical analyses. In experiments with cell lines or in vitro assays with recombinant enzymes, there was no blinding or randomization. In mouse-related studies, the analyst was typically blinded to the experimental group. Generally, experiments were carried out with $n \geq 3$ biological replicates, with the precise number stated throughout the Figure legends.

## Data availability

The proteomics source data have been deposited in the PRIDE database (https://www.ebi.ac.uk/pride/). The dataset for sulforaphane target identifications and proteins phosphorylation identifications are available as PXD061882 and PXD061655, correspondingly.

The source data of this paper are collected in the following database record: biostudies:S-SCDT-10_1038-S44321-025-00267-7.

## Peer review information

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

## Acknowledgements

The authors thank Stuart Bradley at King's College London for technical support and Theracryf plc for providing SFX-01. This work was supported by a project grant from the British Heart Foundation. PE is supported by The Barts Charity Cardiovascular Programme Award G00913 and by project or program grants from the Medical Research Council and the UK Research and Innovation. The authors acknowledge the metabolic flux analysis facility of the Barts School of Medicine and Dentistry, created with the support of the Barts and the London Charity—grant number MGU0401. EL is supported through a National Institute for Health and Care Research Academic Clinical Lecturer scheme, and AJM through a Cancer Research UK Senior Fellowship. The histology facility is a part of Barts Cancer Institute and is supported by CR-UK. The mass spectrometry facility at Barts Cancer Institute is supported by CR-UK (Core award - C16420/A18066).

## Author contributions

**Hyun-Ju Cho**: Conceptualization; Formal analysis; Investigation; Methodology; Writing—original draft; Writing—review and editing. **Joy Smith**: Formal analysis; Investigation; Project administration. **Christopher H Switzer**:

Conceptualization; Formal analysis; Investigation; Writing—original draft; Writing—review and editing. **Eleni Louka**: Conceptualization; Formal analysis; Investigation; Writing—original draft. **Rebecca L Charles**: Investigation. **Oleksandra Prysyazhna**: Formal analysis; Investigation. **Ewald Schroder**: Formal analysis; Investigation. **Mariana Fernandez-Caggiano**: Investigation; Methodology. **Daniel Simoes de Jesus**: Investigation. **Seda Eminaga**: Formal analysis; Investigation; Methodology. **Xiaoke Yin**: Formal analysis; Investigation. **Xiaoping Yang**: Formal analysis; Investigation. **Steven Lynham**: Formal analysis; Funding acquisition; Investigation; Writing—original draft; Project administration. **Manuel Mayr**: Formal analysis; Funding acquisition; Investigation; Writing—original draft; Project administration; Writing—review and editing. **Valle Morales**: Formal analysis; Investigation; Writing—review and editing. **Katiuscia Bianchi**: Formal analysis; Funding acquisition; Investigation; Project administration; Writing—review and editing. **Vinothini Rajeeve**: Data curation; Formal analysis; Investigation; Methodology; Writing—review and editing. **Pedro R Cutillas**: Data curation; Supervision; Methodology; Writing—review and editing. **Adam J Mead**: Conceptualization; Formal analysis; Funding acquisition; Writing—original draft; Project administration; Writing—review and editing. **Philip Eaton**: Conceptualization; Formal analysis; Funding acquisition; Writing—original draft; Project administration; Writing—review and editing.

Source data underlying figure panels in this paper may have individual authorship assigned. Where available, figure panel/source data authorship is listed in the following database record: biostudies:S-SCDT-10_1038-S44321-025-00267-7.

## Disclosure and competing interests statement

The authors declare no competing interests. PE acknowledges he was an unpaid advisor to Theracryf plc.

# Expanded View Figures

## Expanded View Figure

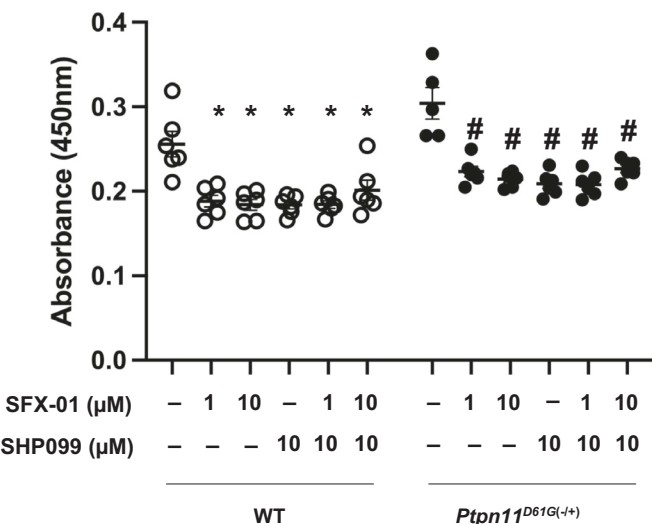

**Figure EV1. Combined treatment with SFX-01 and Shp2 did not accentuate their individual attenuation of proliferation.**

CD11b+ bone marrow cells isolated from WT and *Ptpn11*[D61G(−/+)] mice treated with SFX-01, SHP099 or both together. SFX-01 or SHP099 attenuated cell proliferation and this was not accentuated when these interventions were combined. Data are presented as means ( ± SEM; $n = 5–6$) with *P* values calculated by two-way ANOVA with Tukey's multiple comparison test. *$P < 0.00001$ versus untreated WT or #$P < 0.00001$ versus untreated *Ptpn11*[D61G(−/+)].

