## [Peer Review File · EMBO Molecular Medicine]

SFX-01 is therapeutic against myeloproliferative disorders caused by activating mutations in Shp2

Hyun-Ju Cho, Joy Smith, Christopher Switzer, Eleni Louka, Rebecca Charles, Oleksandra Pryszyzhna, Ewald Schroder, Mariana Fernandez-Caggiano, Daniel de Jesus, Seda Eminaga, Xiaoping Yang, Xiaoke Yin, Steven Lynham, Manuel Mayr, Valle Morales, Katuscia Bianchi, Vinothini Rajeeve, Pedro Cutillas, Adam Mead, and Philip Eaton

Corresponding author: Philip Eaton (p.eaton@qmul.ac.uk)

Review Timeline:

Submission Date:	4th Apr 23
Editorial Decision:	25th Apr 23
Revision Received:	17th Mar 25
Editorial Decision:	8th Apr 25
Revision Received:	16th May 25
Editorial Decision:	6th Jun 25
Revision Received:	19th Jun 25
Accepted:	24th Jun 25

Editor: Lise Roth

Transaction Report:

25th Apr 2023

Dear Prof. Eaton,

Thank you for the submission of your manuscript to EMBO Molecular Medicine. We have now heard back from the three referees who agreed to evaluate your manuscript. As you will see below, the reviewers raise substantial concerns on your work, which unfortunately preclude its publication in EMBO Molecular Medicine in its current form.

The reviewers find that the question addressed by the study is of potential interest, however they remain unconvinced that some of the major conclusions are sufficiently supported by the data. They thus raise the following major issues:

- lack of demonstrated causality
- unspecific nature of SFX-01
- absence of toxicity study.

Addressing the reviewers concerns in full (above points as well as other reviewers' comments) will be necessary for further considering the manuscript in our journal. As revising the manuscript according to the referees' recommendations appears to require a lot of additional work and experimentation, and given the potential interest of your findings, we are ready to extend the deadline to 6 months with the understanding that acceptance of the manuscript would entail a second round of review.

EMBO Molecular Medicine encourages a single round of revision only and therefore, acceptance or rejection of the manuscript will depend on the completeness of your responses included in the next, final version of the manuscript. For this reason, and to save you from any frustrations in the end, I would strongly advise against returning an incomplete revision. Should you find that the requested revisions are not feasible within the constraints outlined here and prefer, therefore, to submit your paper elsewhere, we would welcome a message to this effect.

We require:

4) A .docx formatted letter INCLUDING the reviewers' reports and your detailed point-by-point responses to their comments. As part of the EMBO Press transparent editorial process, the point-by-point response is part of the Review Process File (RPF), which will be published alongside your paper.

5) A complete author checklist, which you can download from our author guidelines (<https://www.embopress.org/page/journal/17574684/authorguide#submissionofrevisions>). Please insert information in the checklist that is also reflected in the manuscript. The completed author checklist will also be part of the RPF.

6) Please note that all corresponding authors are required to supply an ORCID ID for their name upon submission of a revised manuscript.

7) It is mandatory to include a 'Data Availability' section after the Materials and Methods. Before submitting your revision, primary datasets produced in this study need to be deposited in an appropriate public database, and the accession numbers and database listed under 'Data Availability'. Please remember to provide a reviewer password if the datasets are not yet public (see <https://www.embopress.org/page/journal/17574684/authorguide#dataavailability>).

In case you have no data that requires deposition in a public database, please state so in this section (This study includes no data deposited in external repositories). Note that the Data Availability Section is restricted to new primary data that are part of this study.

8) For data quantification: please specify the name of the statistical test used to generate error bars and P values, the number (n) of independent experiments (specify technical or biological replicates) underlying each data point and the test used to calculate p-values in each figure legend. The figure legends should contain a basic description of n, P and the test applied. Graphs must include a description of the bars and the error bars (s.d., s.e.m.). Please provide exact p values.

13) Author contributions: CRediT has replaced the traditional author contributions section because it offers a systematic machine readable author contributions format that allows for more effective research assessment. Please remove the Authors Contributions from the manuscript and use the free text boxes beneath each contributing author's name in our system to add specific details on the author's contribution. More information is available in our guide to authors.

16) As part of the EMBO Publications transparent editorial process initiative (see our Editorial at <http://embomolmed.embopress.org/content/2/9/329>), EMBO Molecular Medicine will publish online a Review Process File (RPF) to accompany accepted manuscripts.

In the event of acceptance, this file will be published in conjunction with your paper and will include the anonymous referee reports, your point-by-point response and all pertinent correspondence relating to the manuscript. Let us know whether you agree with the publication of the RPF and as here, if you want to remove or not any figures from it prior to publication. Please note that the Authors checklist will be published at the end of the RPF.

I look forward to receiving your revised manuscript.

Yours sincerely,

Lise Roth

**** Reviewer's comments ****

Referee #1 (Remarks for Author):

In their manuscript entitled "SFX-01 is therapeutic against myeloproliferative disorders caused by activating mutations in SHP2", Smith J and colleagues identified the tyrosine phosphatase SHP2 as one target of the SFX-01 compound. They found that SFX-01, by covalently modifying cysteines close to its catalytic site, inhibits SHP2 but also, importantly, pathological variants of the protein that cause juvenile myelomonocytic leukemia (JMML) and Noonan syndrome (NS). As a consequence, they evaluated whether SFX-01 can have therapeutic properties towards hyperactive SHP-driven myeloproliferative dysfunctions in vitro and in vivo, and found that SFX-01 treatment rescued myeloproliferation in a NS mouse model and inhibits the proliferation of hematopoietic stem cells derived from JMML patients.

The study is well designed, the presented results are convincing in general and well-discussed. However, it remains essentially correlative in its current format and could be strengthened with additional experiments demonstrating a causal link.

1-A key issue is to demonstrate to which extent SFX-01 effects on myeloproliferation go through SHP2 inhibition. The authors started to investigate this question by comparing the effect of SFX-01 and SHP099 on active SHP2 or active KRAS-expressing HEK293 cells. Please note that the corresponding figure (S9) is incomplete, the right part being masked. This experiment did not allow the authors to conclude for a specific effect since both molecules inhibited cell proliferation to the same extent. Maybe a dose dependent combination of the two treatments can better reveal if SFX-01 effects are solely due to SHP2 inhibition (absence of synergistic effect).

Moreover, it could be interesting to benchmark SFX-01 towards SHP099 in the NS mouse model. Although SHP099 may not display strong inhibitory effect on some variants of SHP2 (but seems to work on D61A variant in this study, as suggested by conclusions from figure S9), it may still have a significant effect by reducing the activity of the WT protein pool (i.e. encoded by the WT allele).

3-a more complete characterization of the phenotype of myeloid cells upon SFX-01 treatment would strengthen the presented results: GM-CSF/M-CSF sensitivity, pro-inflammatory profile, immunophenotyping of hematopoietic cells combined to gene expression pattern analysis...

4-if I understand correctly the mechanism of action of SFX-01, SFX-01 adducts are only transient and evolve to a dithiolethione product with inhibitory properties. As C333S, C367S or C333/367S mutants of SHP2 remain SFN-bound, could they be resistant to SFX-01 inhibition? If so they could be used as important controls in JMML cell lines, to assess if these cells retain a proliferative phenotype when expressing such mutant.

5-in the in vivo experiments, NS mice have been treated for 10 weeks with SFX-01. Besides congenital defects, a number of additional features seem to occur at adulthood or to be evolutive throughout life. Could the authors document an improvement of some of the other NS-associated clinical traits (cardiopathy, platelet aggregation, inflammation, glucose intolerance, musculoskeletal defects).

6-many mitochondrial proteins seem to be targeted by SFN/SFX-01, did the authors measure the effect of SFX-01 treatment on mitochondria activity (respiration, activity of specific complex)? Could reduced energy production explain the proliferation arrest?

7- the authors observed that SFX-01 treatment activated ERK1/2. Has this experiment been performed upon agonist stimulation (e.g. EGF)? If not, such long term exposure could also reflect global metabolic changes. Is this ERK1/2 overactivation also observed in tissues from SFX-01-treated NS mice?

8-Beside scientific issues, as a scientific advisor for Evgen Pharma plc, the corresponding author should maybe clarify his potential conflict of interest.

Referee #2 (Comments on Novelty/Model System for Author):

please see the comments to the authors below.

Referee #2 (Remarks for Author):

Comments to the Author:

This manuscript authored by Smith et al. reported the identified protein targets of SFX-01, including Shp2. SFX-01 is an activating mutant Shp2 inhibitor and may offer the beneficial effects in patients with JMML or Noonan syndrome.

However, the manuscript has a number of weaknesses that should be addressed before considering for publication in EBMO Molecular Medicine.

1. First, the authors identified Shp2 as an SFX-01 molecular target, and SFN-modification inhibited Shp2 phosphatase activity therefore SFX-01 restored the myeloid cell number to wild type levels and resulted in cell cycle arrest, which suggested SFX-01 may have therapeutic effects for JMML and other Shp2-driven pathologies. As shown in supplementary table, there are tons of targets could be modified by SFN. Hence, the authors should provide more evidence to convince the therapeutic effects in JMML/Noonan syndrome mouse model are related to Shp2 inhibition rather than other proteins (mitochondrial proteins).

A. The authors need to show the Shp2 activity and SFN-modification alterations in isolated leukocytes in mutant Shp2 mouse model, related to FIG.3 and 4.

B. The authors may perform an unbiased RNA-seq/proteomic analysis to demonstrate Shp2-related pathways would be significantly affected after SFX-01 treatment in mutant Shp2 mouse model. As reported from other researchers, Shp2 regulates downstream signaling events such as cellular proliferation and cell-cycle progression by activating Ras while inhibiting STAT1 signaling. Following the literature, the authors observed STAT1 signaling change upon SFX-01 treatment, how about other canonical Shp2-regulated pathway? The authors may convince us the STAT1 alteration is related to Shp2 inhibition rather than a consequence of SFN modification on multiple targets.

2. Second, the authors claimed SFX-01 is a first-in-class activating mutant Shp2 inhibitor, which may show more benefits than allosteric inhibitors. However, the only comparison experiment in Fig.S9 showed SHP099 even has a better or equal effect on SHP2 (D61A)-overexpressed cells, compared to SFX-01.

A. Although the authors explain a lot regarding this result, it's better to compare their therapeutic effects in Shp2 mutant mouse model.

B. It's surprising the HEK293 cells expressing either active Shp2 (D61A) or active KRas (G12C) didn't show phospho-ERK activation. How to explain these?

Specific comments:

1. The authors have performed amounts of work for this paper. That said, the western blot data in left panel of Fig 5A should be more convincing if the authors increase more mouse number and image quality. Fig 5B should show us total STAT1 expression as well.

2. Why in vitro incubation using same amount of Shp2 (1.6 nM) in Fig 2D showed such a big difference in Shp2 immunoblots image? And the authors should present the molecular weight of cysteinyl-SFN like Fig 2G. In addition, could authors make the rationale about the time-dependent loss of cysteinyl-SFN in lower dosage of SFX-01? Does this dosage show any relevant to in vivo blood drug concentration in vivo?

3. The authors may explain how to quantify the immunoblot grey intensity in Fig S4A. For example, it is obvious that SFX-01 induces more SFN-modification in Ptpn11-mutant mouse in day 10 compared with WT mouse.

4. In the legend of Fig 4E, the authors described "Mean changes in spleen size in WT or Ptpn11D61G(-/+) mice over 10-weeks of SFX-01 treatment compared to control animals". Based on my understanding, it should be "Mean changes in spleen size in WT or Ptpn11D61G(-/+) mice before and after 10-weeks of SFX-01 treatment"?

5. To fit the EMM manuscript request, more details should be written in the legend, including drug dosage and treated time.

6. legend Fig 2D: 0.109 μ M SFN., it should be SFX-01?

7. pg7 line9: As in Fig 1D, which should be Fig 2D?
8. There was lack of scale bar of IHC images in Fig 4B.
9. Method: Immunostaining, "20 mM" rather than "20mM".
10. Method, SFN is stable in water for at least 6-days (Fig. S3), it should be Fig. S5.

Referee #3 (Comments on Novelty/Model System for Author):

Not so sure on quality of the drug due to broad range cysteine modifier

Referee #3 (Remarks for Author):

The study "SFX-01 is therapeutic against myeloproliferative disorders caused by activating mutations in Shp2" is in format of a research article submitted to EMBO MOL MED. As such the work should contain experimental details that can be followed by the reader or reviewer and a clear description for Material and Methods section is available in the Supplementary Data, but more essential information should be moved to main text body since some of it is essential information. Critical data or information is not disclosed or introduced. The study has some interesting aspects and it focuses on orphan diseases at least with NS that do not have targeted drugs and maybe a dominance of childhood diseases that are not in interest of pharma industry, so these points are clearly positive and deserve credit. However, several shortcomings of the study are also noticeable; Drug specificity issues or suitability or off-target effects, toxicity issues will exist and several mechanistic things are quite limited to purified molecules and not much broadly applicable to proteomic data on cellular systems expressing SHP2 mutations relevant to Noonan Syndrome or MPN to be in main focus for targeting. As the manuscript was provided the reviewer suggests that it is not suited for publication in current format, but editing better and a number of experiments and controls can be performed for major revision to make it more round or better controlled for drug specificity issues on a broad range cysteine modifier and if that is really a good way to go. As it is, one cannot conclude on suitability of the drug in focused disease context or compared to other even less specific cysteine modifiers. If toxicity is in their model really not happening needs experimental validation and drug was not applied in dose-dependent evaluation, another bottleneck, but given in water ad libitum. Moreover, the text body and intro or result section lack detailed information as outlined in the following.

Major:

1) The discussion is in general quite limited and should more critically discuss known negative side effects and the possible applications so far of SFX-01 and it is not true that many clinical trials are done with it, only two are found completed on clinicaltrials.gov site from NIH which is USA government operated. Thus, the clinicaltrials.gov site is pretty comprehensive and only two studies listed with SFX-01 is limited and these are completed, one phase 2 in breast cancer as combi therapy, the other study very unrelated and irrelevant eventually, at least in regard to reviewer opinion. Other than this several reports suggest it has broad range neurological disease action as a broad range acting drug due to covalent nucleophilic attack of cysteine residues. Here, it should be clearly introduced that all PTPs and not only SHP2 have catalytic Cysteine sites, so how about CD45 as the most expressed membrane tyrosine phosphatase in all blood cells except the erythroid lineage. Is CD45 also found to be pulled down in myeloid blasts? Can be a control blot and investigation. SFX-01 looks like a very unspecific broad range acting drug as a covalent cysteine modifier, maybe not as broad e.g. as "Stattic" a wrongly claimed STAT3 inhibitor, that might serve as a good model substance to compare it with, explained below in a separate point since that would allow a better conclusion for drug specificity issues and impact. At the end the authors claim that SFCX-01 is a good drug, but a comparison e.g. to Stattic which is a low-activity, non-peptide small molecule which turned out to be totally unspecific modifying any cysteine residue exposed on the surface of molecules could be a suitable assay if SFX-01 is more specific, then good, if not then the reviewer has strong doubts on toxicity and off-target effects. No toxicity testing in mice of their drinking water treatment was done, but this is mandatory. SFX-01 when dosed high was reported to have GI tract toxicity, but this was not analyzed, sections of GI tract, liver and kidney damage parameter, hematocytometry should be included, particularly due to placing the drug into drinking water which can be called uncontrolled or highly dosed? Mice can drink quite variable particularly if they have or have not disease, where NS model is quite harsh, but do the animals tolerate that compared to control group needs to be disclosed. Stattic was also published in a number of articles as a non selective STAT3 blocker or a broad range acting drug with questionable conclusion. The reviewer thinks the story would benefit on a comparison on some critical protein targets of SFX-01 and Stattic, SHP2, STAT1, STAT3 and STAT5B at least from expression and activity status and a couple of Western blots or proteomic could be runned to get an idea of a correct and state-of-the-art pharmacologic broad or more specific range cysteine modifier. Is SFX-01 really better than Stattic and if so why?

2) Authors should not only check on STAT1 and pYSTAT1 levels, they should also include and conclude on the analysis for the pY-STAT3 and pY-STAT5A/pYSTAT5B and total STAT3 as well as total STAT5A or STAT5B levels in their model cell systems, since covalent cysteine modifications are known actions on small molecular weight covalent binders of STAT molecules. STAT family members do also interact with SHP2 as it was also mapped in several papers and that can be cited. D type cyclins are in particular in myeloid or other acute leukemia or MPN targets of STAT3 and/or STAT5 action and they can be suppressed via

STAT1 tumor suppressor induction. Thus, extended protein assays should monitor STAT3, STAT5A and STAT5B expression and activity status, particularly since authors only focus on STAT1 action associated with interferon signaling largely. RNA-seq pathway analysis with and without treatment in diseased mouse context should better shed light on core cancer pathway activity. 3) RNA-seq analysis with and without drug would allow a better conclusion on pathway analysis hit by SFX-01 versus vehicle control in drinking water of mice of isolated myeloid blasts that are e.g. FACS sorted and purified and sequenced in bulk as a good enough analysis in three experimental mice each group to conclude on targeting aspects. Here, a good surface molecule target of STAT1 action should be MHC class I surface expression which can be monitored in parallel as well. The consequence of it might be foreign neoantigen expression, but that could be discussed if changed. Thus, authors could look by FACS on HLA subtype expression level on their mice which are most likely inbred pure background strains, which is another important information for readers to be disclosed. MHC class I should be upregulated upon their drug treatment. Other direct STAT1 targets are caspase induction or p21 cell cycle inhibitor induction or interferon regulatory factor induction or antiviral genes, where Mx1 GTPase are good targets to be e.g. eluted or to more broadly see and discuss interferon target genes. Unbiased RNA-seq would reveal better insight. Thus, a more broader anti-survival, anti-proliferative or anti-inflammatory action could originate from a broader impact on JAK-STAT-SOCS protein components, known to be essential genes in MPN and acute leukemias of the myeloid lineages.

4) Normally text editing is minor, but since quite a lot of info in this manuscript is missing the reviewer opens here another point as major for lack of specific information to better grasp the study or to understand the nature of the model, the expression, etc....As the manuscript was written one does not have the impression that details might be so clear why e.g. cardiac tissue from these mice was used for analysis when the headline states "myeloproliferative disorder", but it is clear when reading up on these mice that the cardiac tissue is affected in half of the mice and how the transgenic model was generated should be stated as well in detail, e.g. is it a knockin, is it a transgene that is randomly integrated, that matters at the end to understand the model, it expresses one SHP2 variant found in Noonan syndrome, etc.... Thus, advice is to write that clearly in with own words into the introduction or into the result section even for mouse model description, essential information is otherwise lost. Thus, the reviewer feels that the mouse model info is crucial for judgment and it makes the reviewing process faster if such information would be provided, since the reader/reviewer has then not to look that up and also most readers will not understand that based on background of the wealth of mouse models available today and differences in model character for expression, tissue type expression, activation status due to mutation, etc. The description by the Ben Neel lab on the Noonan mouse model carrying the SHP2 variant is very detailed in the abstract and why not using that take home info into the manuscript. Here, it is quoted from the abstract in following lines: "... mice expressing the Noonan syndrome-associated mutant D61G. When homozygous, the D61G mutant is embryonic lethal, whereas heterozygotes have decreased viability. Surviving Ptpn11D61G/+ embryos (~50%) have short stature, craniofacial abnormalities similar to those in Noonan syndrome, and myeloproliferative disease. Severely affected Ptpn11D61G/+ embryos (~50%) have multiple cardiac defects similar to those in mice lacking the Ras-GAP protein neurofibromin. Their endocardial cushions have increased Erk activation, but Erk hyperactivation is cell and pathway specific...."

Minor:

5) Along the lines on missing information also the structure and Mw, etc. should be given as well as the R- and L-forms for enantiomer should be disclosed. Evgen Pharma data disclosed that MW: 173.19 + 972.846 g/mol is within suitable range of Lipinski Rule of 5 (<http://www.scfbio-iitd.res.in/software/drugdesign/lipinski.jsp>), which could be discussed as well by the authors to claim it has drug like properties that are good.

Summary of main changes based on feedback from Editor after initial review**Toxicity studies**

We conducted toxicity studies using WT and NS model mice treated with SFX-01, which showed no differences between genotypes. This aligns with previous studies and FDA approval for human trials, confirming that SFX-01 is safe. This extensive 10-week study involved multiple endpoint measurements and generated a robust dataset from 50 mice (Appendix Figure S6), requiring significant time and resource allocation.

Phosphoproteomics analysis

We performed phosphoproteomics analysis on CD11b+ myeloid bone marrow cells isolated from WT and NS mice, with or without SFX-01 treatment. To achieve deep phosphoproteome coverage, we isolated CD11b+ myeloid bone marrow cells from 54 mice, which was both resource- and time-intensive. However, the results proved highly valuable because this unbiased approach showed SFX-01 increased Shp2 substrate phosphorylation consistent with inhibition of its phosphatase activity (Figure 5G). Furthermore, pathway enrichment analysis of these data showed that SFX-01 attenuated pathways associated with cell proliferation (Appendix Figure S11).

Shp2 inhibition studies with SFX-01 and SHP099 individually or together

As per guidance during the last review, we conducted additional experiments to assess whether the effects of SFX-01 are mediated, at least in part, through Shp2 inhibition by co-treating with SHP099. These studies demonstrated the attenuation of proliferation achieved individually by SFX-01 or SHP099 is not accentuated when these interventions are combined (Appendix Figure S14). This observation reinforces that SFX-01 attenuates proliferation via its inhibitory action on Shp2 because if it was significantly mediated by alternate mechanisms then an additive effect is to be anticipated when the gold standard SHP099 inhibitor was also present - which was not observed.

SFX-01 and developmental outcomes

Reviewers also inquired about the potential of SFX-01 to mitigate other NS phenotypes, such as cardiac or musculoskeletal defects, which develop *in utero*. To investigate this, we conducted extensive studies in which pregnant dams were treated with SFX-01 to assess whether inhibition of SHP2 phosphatase activity could enhance the birth rate of heterozygous offspring or enable homozygous NS offspring to survive to birth - an event that does not otherwise occur. However, unexpectedly, SFX-01 exposure during development not only failed to produce viable homozygous NS offspring but also led to the complete loss of heterozygous mice, which are normally viable. This outcome was consistent across two separate studies: one in which SFX-01 was administered from pre-conception through pregnancy and another in which it was given only post-gastrulation (Appendix Figure S9).

Next, we provide a point-by-point response to the reviewers' comments in **red**:

Referee #1 (Remarks for Author):

In their manuscript entitled "SFX-01 is therapeutic against myeloproliferative disorders caused by activating mutations in SHP2", Smith J and colleagues identified the tyrosine phosphatase SHP2 as one target of the SFX-01 compound. They found that SFX-01, by covalently modifying cysteines close to its catalytic site, inhibits SHP2 but also, importantly, pathological variants of the protein that cause juvenile myelomonocytic leukemia (JMML) and Noonan syndrome (NS). As a consequence, they evaluated whether SFX-01 can have therapeutic properties towards hyperactive SHP-driven

myeloproliferative dysfunctions in vitro and in vivo, and found that SFX-01 treatment rescued myeloproliferation in a NS mouse model and inhibits the proliferation of hematopoietic stem cells derived from JMML patients.

The study is well designed, the presented results are convincing in general and well-discussed. However, it remains essentially correlative in its current format and could be strengthened with additional experiments demonstrating a causal link.

The issue about causality is critical and arguably the most important point raised by the reviewers.

We think that on balance that we provided reasonable evidence that inhibition of Shp2 by SFX-01 is causally therapeutic against myeloproliferative disease. For example, we unbiasedly detected using proteomics that SFX-01 adducts to Shp2 and follow up studies showed this inhibits the Shp2 phosphatase activity - as it also does in our comprehensive *in vitro* studies. This adductive inhibition was therapeutic, correcting the myeloproliferation in a mouse model of NS and in isolated primary JMML patient cells that each have mutations that activate Shp2 above wild type levels. Traditionally, this would be considered good evidence of how the drug exerts its therapeutic actions, but at the same time the community would be aware that the molecule will hit other targets, and those actions may potentially contribute to the drug action. Concomitant correction of a correcting a disease-causing enzyme activity and disease phenotype would normally speak to a causal relationship. We think causality has become a particular focus here because we have defined multiple targets SFX-01 that it modifies. In reality, all molecules will hit many targets, but because we have defined them, we have created this issue.

We included a Discussion section entitled "SFX-01 is not fully selective for Shp2" that considers this very point. It is perhaps helpful to remember that all drugs will have off target effects, but routinely identifying targets of conventional, non-covalent drugs such as SHP099 that reversibly binds Shp2 (and other proteins) is not readily achievable because of their transient interaction. Nevertheless, such interactions will take place, and some are likely to contribute to the biological actions of these drugs. Whilst the Shp2 inhibitor SHP099 is considered highly selective, in reality the multitude of targets that it likely binds are not known, and so off-target effects cannot be readily assessed. Although we have defined many targets modified when a mouse consumes SFX-01, it is perhaps unfair to hold us to a standard that is not typically applied to conventional drugs where the full list of targets is elusive. The crucial point is that SFX-01 inhibits Shp2, and our pre-clinical studies show it corrects aberrant myeloid cell growth caused by mutations that activate this phosphatase. We acknowledge in the revised manuscript that SFX-01 likely has poly-pharmacological actions but justifiably conclude a significant mode of action is inhibition of Shp2. In addition, and we should remember that SFX-01 has passed a human safety evaluation and has and continues to be tested in clinical trials. Notably, over-the-counter drugs omeprazole and acetaminophen target H⁺/K⁺ATPase and cyclooxygenase respectively are covalent modifiers as is SFX-01. Consequently, antibodies to those drugs could be made that allowed comprehensive target identification as we have for SFX-01. Below are example western blots from cells or mice exposed to omeprazole or acetaminophen. Clearly, many targets are modified, reminiscent of what we found with SFX-01.

Figures for reviewers removed.

Nevertheless, we performed additional experiments aimed at further assessing causality as suggested by the Reviewer. As described in our comments on the first page, phosphoproteomics analysis assessing changes in phosphorylation upon exposure to SFX-01 showed that it induced changes in phosphorylation consistent with inhibition of Shp2 as included in this revision (Figures 5G, and Appendix Figure S11).

1-A key issue is to demonstrate to which extent SFX-01 effects on myeloproliferation go through SHP2 inhibition. The authors started to investigate this question by comparing the effect of SFX-01 and SHP099 on active SHP2 or active KRAS-expressing HEK293 cells. Please note that the corresponding figure (S9) is incomplete, the right part being masked. This experiment did not allow the authors to conclude for a specific effect since both molecules inhibited cell proliferation to the same extent. Maybe a dose dependent combination of the two treatments can better reveal if SFX-01 effects are solely due to SHP2 inhibition (absence of synergistic effect).

A complete and revised Appendix Figure S12 (Supplementary Fig. 9 has been updated to Appendix Figure S12) is now included. As suggested, we conducted additional experiments to assess whether the effects of SFX-01 are mediated, at least in part, through Shp2 inhibition by co-treating with SHP099. These studies demonstrated the attenuation of proliferation achieved individually by SFX-01 or SHP099 is not accentuated when these interventions are combined (see Appendix Figure S14). This observation reinforces that SFX-01 attenuates proliferation via its inhibitory action on Shp2 because if it was significantly mediated by alternate mechanisms then an additive effect is to be anticipated when the gold standard SHP099 inhibitor was also present, which was not observed. Again, these data strengthen the causal link highlighted by the Reviewer.

Moreover, it could be interesting to benchmark SFX-01 towards SHP099 in the NS mouse model. Although SHP099 may not display strong inhibitory effect on some variants of SHP2 (but seems to

work on D61A variant in this study, as suggested by conclusions from figure S9), it may still have a significant effect by reducing the activity of the WT protein pool (i.e. encoded by the WT allele).

As our resources are limited, we did not perform *in vivo* SHP099 studies. However, we have undertaken a number of additional, large studies that are included in this revision.

3-a more complete characterization of the phenotype of myeloid cells upon SFX-01 treatment would strengthen the presented results: GM-CSF/M-CSF sensitivity, pro-inflammatory profile, immunophenotyping of hematopoietic cells combined to gene expression pattern analysis...

Again, because of limited resources we did not perform these studies. We think that the new data included in the revision strengthens the evidence that SFX-01 inhibits Shp2 to limit myeloproliferation in NS.

4-if I understand correctly the mechanism of action of SFX-01, SFX-01 adducts are only transient and evolve to a dithiolethione product with inhibitory properties. As C333S, C367S or C333/367S mutants of SHP2 remain SFN-bound, could they be resistant to SFX-01 inhibition? If so they could be used as important controls in JMML cell lines, to assess if these cells retain a proliferative phenotype when expressing such mutant.

We understand the general concept behind the proposed experiments, but the mutants mentioned will not prevent adduction and inhibition of Shp2 by SFX-01. This is because the adduction of the electrophile to the catalytic cysteine is itself alone sufficient to inhibit phosphatase activity because it requires a free -SH. Eventually the adduct transitions to the dithiolethione in WT Shp2, which is also inhibitory because again this renders the -SH unavailable for catalysis. Mutation of the oxidant sensitive regulatory thiol is a common strategy in redox enzymes studies, but this is not helpful in this scenario as mutation of that cysteine renders the Shp2 inactive. Thus, unfortunately, we cannot remove the effect of the drug (i.e., Shp2 inhibition) by making the cysteine mutant.

The Reviewer also mentioned JMML cell lines, and we wanted to clarify that these are primary cultures from human JMML patients. They cannot be passaged many times because they lose their JMML phenotype, and this means only very limited studies can be performed with these precious and very limited cells.

5-in the *in vivo* experiments, NS mice have been treated for 10 weeks with SFX-01. Besides congenital defects, a number of additional features seem to occur at adulthood or to be evolutive throughout life. Could the authors document an improvement of some of the other NS-associated clinical traits (cardiopathy, platelet aggregation, inflammation, glucose intolerance, musculoskeletal defects).

We recognise the importance of this question about SFX-01 rescuing other pathogenic features of the NS mice. A major consideration is that many of these (e.g., musculoskeletal or cardiac valve defects) are patterned *in utero* or are not present in this mouse model (e.g., cardiac hypertrophy or platelet aggregation). The original publication of the D61G Shp2 mice (PMID: 15273746) says they have "normal hematocrit and platelet counts" and "cardiac hypertrophy was not observed". We realise other NS models do have such phenotypes. However, as this is an important point, we tried to address this by administering SFX-01 to mice *in utero* by feeding the mothers during pregnancy using two comprehensive studies. Pregnant dams were treated with SFX-01 to assess whether inhibition of SHP2 phosphatase activity enhanced the birth rate of heterozygous offspring or enabled homozygous NS offspring to survive to birth. However, unexpectedly, SFX-01 exposure during

development not only failed to produce viable homozygous NS offspring but also led to the complete loss of heterozygous mice, which are normally viable. This outcome was consistent across two separate studies: one in which SFX-01 was administered from pre-conception through pregnancy and another in which it was given only post-gastrulation as shown in Appendix Figure S9.

6-many mitochondrial proteins seem to be targeted by SFN/SFX-01, did the authors measure the effect of SFX-01 treatment on mitochondria activity (respiration, activity of specific complex)? Could reduced energy production explain the proliferation arrest?

We performed mitochondrial function tests using a Seahorse Analyzer to ascertain the impact, if any, of SFX-01 on mitochondrial function - we also included studies with SHP099. Appendix Figure S13 shows that 3 or 10 μ M SFX-10 or SHP099 did not significantly alter mitochondrial function as indexed by oxygen consumption rate (OCR). We therefore conclude that SFX-01 is not exerting its therapeutic actions through altering mitochondrial function.

7- the authors observed that SFX-01 treatment activated ERK1/2. Has this experiment been performed upon agonist stimulation (e.g. EGF)? If not, such long term exposure could also reflect global metabolic changes. Is this ERK1/2 overactivation also observed in tissues from SFX-01-treated NS mice?

We have not measured pERK1/2 after treatment with agonists such as growth factors. However, we find that pERK is commonly increased in cells or mice exposed to SFX-01. This is shown in Appendix Figure S12C as was originally included, but we also now include additional data in Appendix Figure S12D that shows that pERK was increased in cells or mice exposed to SFX-01. Furthermore, our newly included unbiased phosphoproteomics study shows that both WT and NS CD11b+ myeloid bone marrow cells exposed to SFX-01 increased pERK levels (Appendix Figure S12E, F). It's evident across multiple datasets that SFX-01 increases pERK despite its ability to decrease proliferation.

ERK phosphorylation data (Appendix Figure S12C) was from HEK293 cells cultured with or without SFX-01 for 96 hours in DMEM media + 10% FBS. We will now analyse and report the impact of SFX-01 on pERK levels in mice *in vivo* as requested (Appendix Figure S12D).

8-Beside scientific issues, as a scientific advisor for Evgen Pharma plc, the corresponding author should maybe clarify his potential conflict of interest.

The role of Eaton as an advisor is acknowledged in the manuscript now, as is the fact that this role is fulfilled gratis.

Referee #2 (Comments on Novelty/Model System for Author):

This manuscript authored by Smith et al. reported the identified protein targets of SFX-01, including Shp2. SFX-01 is an activating mutant Shp2 inhibitor and may offer the beneficial effects in patients with JMML or Noonan syndrome.

However, the manuscript has a number of weaknesses that should be addressed before considering for publication in EBMO Molecular Medicine.

1. First, the authors identified Shp2 as an SFX-01 molecular target, and SFN-modification inhibited Shp2 phosphatase activity therefore SFX-01 restored the myeloid cell number to wild type levels and

resulted in cell cycle arrest, which suggested SFX-01 may have therapeutic effects for JMML and other Shp2-driven pathologies. As shown in supplementary table, there are tons of targets could be modified by SFN. Hence, the authors should provide more evidence to convince the therapeutic effects in JMML/Noonan syndrome mouse model are related to Shp2 inhibition rather than other proteins (mitochondrial proteins).

Please see our reply to Reviewer 1 about the difficulties of definitively proving that a drug is therapeutic by modulating a single target. To reiterate, we included Discussion text acknowledging the likely role of poly-pharmacology involving targets other than Shp2. In reality, it is exceedingly difficult to prove definitively a drug operates by solely modulating a single target, even if one drug - one target was a credible prospect. Put another way, how can we assess the importance of Shp2 inhibition when other targets are not known. Or even when they are, there is difficulty in making comparisons of their relative contributions. We know that SFX-01 inhibits Shp2 *in vitro*, and it also does this in mice and in cell models with mutant hyperactive Shp2 and this intervention corrects their aberrant hyper-proliferative growth phenotype. We think therefore it is reasonable to conclude this is significantly mediated by SFX-01 inhibiting Shp2.

Identifying and proving a sole molecular target of a drug is a nearly intractable problem, as how can one be sure that other (often unknown) cellular targets are not involved? Of course, there is the theoretical concept that if a putative Shp2 inhibitor (here SFX-01) still impairs growth induced by active Ras, it cannot be via Shp2 inhibition because it is upstream of Ras. As discussed in the manuscript, this very strategy was used to 'prove' SHP099 was highly specific for Shp2 (Chen *et al.*, Nature 2016). Unfortunately, we simply did not obtain the same result, and included this Discussion text: "*While this appears inconsistent with Chen et al.⁹, it is notable they did not formally show the attenuated proliferation caused by SHP099 was rescued by expression of active Ras. Indeed, we note studies that demonstrated some mutant KRas cell models are sensitive to SHP099⁴², especially when grown as 3D multicellular preparations⁴³. Indeed, KRas driven proliferation can be dependent on Shp2^{42,44} and so there is complexity in definitively proving that either of these drugs are solely through inhibition of this phosphatase*". We hope this adds some reasonable considerations about the practical reality of addressing this important issue. Had we not showed there are multiple targets of SFX-01, perhaps we would not been in this predicament, but the reality is all drugs have multiple targets – just that they normally are not comprehensively defined.

Importantly, we have now strengthened the evidence that SFX-01 dependent inhibition of Shp2 causally, at least in part, mediates protection against myeloproliferation in the NS model mice. These newly included studies are described above and relate to a phosphoproteomic analysis demonstrating SFX-01 increases phosphorylation of established Shp2 substrates (see Figure 5G). Additionally, we showed that the reduction in proliferation achieved individually by SFX-01 or SHP099 is not enhanced when both interventions are combined (see Appendix Figure S14). This finding supports the conclusion that SFX-01 attenuates proliferation significantly through its inhibitory action on Shp2. If alternative mechanisms played a significant role, an additive effect would be expected when the gold-standard SHP099 inhibitor was also present—an outcome that was not observed.

We also conducted function tests using a Seahorse Analyzer to assess the potential impact of SFX-01 on mitochondrial activity, including comparative studies with SHP099. As shown in Appendix Figure S13, neither 3 μ M nor 10 μ M of SFX-01 or SHP099 significantly affected mitochondrial function, as indicated by oxygen consumption rate (OCR). These findings indicate that SFX-01 are unlikely to exert its therapeutic actions by altering mitochondrial function.

A. The authors need to show the Shp2 activity and SFN-modification alterations in isolated leukocytes in mutant Shp2 mouse model, related to FIG.3 and 4.

We originally tried to do what the Reviewer suggests, but after sorting the leukocytes and immunoprecipitating Shp2, we were below the detection limits of the assay. Furthermore, the isolation procedure is time consuming and stressful and so there will be complicated changes in signalling that would add complexity. Consequently, we performed phosphoproteomics, as we will come to next.

B. The authors may perform an unbiased RNA-seq/proteomic analysis to demonstrate Shp2-related pathways would be significantly affected after SFX-01 treatment in mutant Shp2 mouse model. As reported from other researchers, Shp2 regulates downstream signaling events such as cellular proliferation and cell-cycle progression by activating Ras while inhibiting STAT1 signaling. Following the literature, the authors observed STAT1 signaling change upon SFX-01 treatment, how about other canonical Shp2-regulated pathway? The authors may convince us the STAT1 alteration is related to Shp2 inhibition rather than a consequence of SFN modification on multiple targets.

We did not have the resource to perform both RNA-seq as well as proteomic analysis, so chose the latter in the form of an unbiased phosphoproteomics study. This was because increased Shp2 substrate phosphorylation can rationally be anticipated when SFX-01 is administered if it inhibits this phosphatase. We performed phosphoproteomics analysis on CD11b+ bone marrow cells isolated from WT and NS mice, with and without SFX-01 treatment. To achieve deep phosphoproteome coverage, we isolated CD11b+ cells from 54 mice, which was both resource- and time-intensive. We could not treat mice with or without the SFX-01 and then isolate the relevant cells as the time-consuming isolation procedure disrupts the signalling, so we used the cell model described. The results proved highly valuable because this unbiased analysis showed SFX-01 increased Shp2 substrate phosphorylation consistent with inhibition of its phosphatase activity (Figure 5G). Furthermore, pathway enrichment analysis of these data showed that SFX-01 attenuated mechanisms associated with cell proliferation (Appendix Figure S11).

2. Second, the authors claimed SFX-01 is a first-in-class activating mutant Shp2 inhibitor, which may show more benefits than allosteric inhibitors. However, the only comparison experiment in Fig.S9 showed SHP099 even has a better or equal effect on SHP2 (D61A)-overexpressed cells, compared to SFX-01.

In retrospect we perhaps should not have claimed “first-in-class” and will tone our conclusions down. Our Appendix Figure S12 (Supplementary Fig. 9 has been updated to Appendix Figure S12) data was included to show our finding that SHP099 likely has off-target effects, and its anti-proliferative actions likely also involve poly-pharmacology. This is unlikely to be due to SHP099 inhibiting the open-conformation of Shp2, because Sun *et al.* and LaRochelle *et al.* have independently shown that SHP099 is less effective against hyper-active forms of Shp2 as we cited in the manuscript.

A. Although the authors explain a lot regarding this result, it's better to compare their therapeutic effects in Shp2 mutant mouse model.

Due to limited resource, we did not perform *in vivo* SHP099 studies, especially as it is not the focus of this study. Thus, we have not prioritised this study but have included others suggested in which the impact of SFX-01 and SHP099 individually or together on proliferation was investigated. The

attenuation of proliferation achieved individually by SFX-01 or SHP099 was not accentuated when these interventions are combined (Appendix Figure S14). This observation reinforces that SFX-01 attenuates proliferation via its inhibitory action on Shp2 because if it was significantly mediated by alternate mechanisms then an additive effect is to be anticipated when the gold standard SHP099 inhibitor was also present, which was not observed.

B. It's surprising the HEK293 cells expressing either active Shp2 (D61A) or active KRas (G12C) didn't show phospho-ERK activation. How to explain these?

The western blot data in Appendix Figure S12C were obtained from cells after 96 hours when those that were untreated were confluent. Therefore, we interpret the lack of phospho-ERK as the result of non-replicating/quiescent cells. This is contrasted with SFX-01 treated cells, which have not achieved confluence (Appendix Figure S12A-B) and therefore ERK activation is possible. Furthermore, we have shown that SFX-01 results in ERK phosphorylation. Therefore, due to the action of SFX-01 and the sub-confluency of these cells at this time point, this may explain the dramatic differences in ERK activation. At earlier time points, mutant Ras and Shp2 do increase ERK phosphorylation. pERK was generically increased in cells or mice exposed to SFX-01. This is shown in Figures Appendix Figure S12C as was originally included, but we also now include additional data (Appendix Figure S12D) that shows that pERK was increased in cells or mice exposed to SFX-01. Furthermore, our newly included unbiased phosphoproteomics study shows that both WT and NS CD11b+ cells exposed to SFX-01 increased pERK levels (Appendix Figure S12E-F). It's evident across multiple datasets that SFX-01 increases pERK despite its ability to decrease proliferation.

Specific comments:

1. The authors have performed amounts of work for this paper. That said, the western blot data in left panel of Fig 5A should be more convincing if the authors increase more mouse number and image quality. Fig 5B should show us total STAT1 expression as well.

The original samples were too old or in short supply to reanalyse for total STAT1. As mentioned above we performed phosphoproteomics and found that SFX-01 significantly increases STAT1 phosphorylation as shown in Figure 5G.

2. Why in vitro incubation using same amount of Shp2 (1.6 nM) in Fig 2D showed such a big difference in Shp2 immunoblots image? And the authors should present the molecular weight of cysteinyl-SFN like Fig 2G. In addition, could authors make the rationale about the time-dependent loss of cysteinyl-SFN in lower dosage of SFX-01? Does this dosage show any relevant to in vivo blood drug concentration in vivo?

This is discussed in the manuscript; lower dose SFX results in only one cysteine adduct forming, which can resolve to dithiolethione and loss of the SFN-adduct. However, higher SFX causes widespread thiol alkylation, stabilising the SFN adduct because the adjacent cysteine is also adducted by the electrophile when abundant, so preventing dithiolethione formation. It is difficult to relate these cellular studies to what happens *in vivo*, especially as dithiolethione formation is not simply a concentration-dependent phenomenon, but time is crucial too.

3. The authors may explain how to quantify the immunoblot grey intensity in Fig S4A. For example, it is obvious that SFX-01 induces more SFN-modification in Ptpn11-mutant mouse in day 10 compared with WT mouse.

The difference mentioned is due to biological variation, as shown in the graph to the right in Appendix Figure S4A. Overall there is no significant difference between WT and the *Ptpn11*^{D61G(-/+)} mouse, just variation in the protein labelling induced by SFX-01 in an *in vivo* situation.

4. In the legend of Fig 4E, the authors described "Mean changes in spleen size in WT or *Ptpn11*D61G(-/+) mice over 10-weeks of SFX-01 treatment compared to control animals". Based on my understanding, it should be "Mean changes in spleen size in WT or *Ptpn11*D61G(-/+) mice before and after 10-weeks of SFX-01 treatment"?

Yes, this is correct, and the text will be edited.

5. To fit the EMM manuscript request, more details should be written in the legend, including drug dosage and treated time.

We will correct this.

6. legend Fig 2D: 0.109 μ M SFN., it should be SFX-01?

We will correct this.

7. pg7 line9: As in Fig 1D, which should be Fig 2D?

We will correct this.

8. There was lack of scale bar of IHC images in Fig 4B.

We will correct this.

9. Method: Immunostaining, "20 mM" rather than "20mM".

We will correct this.

10. Method, SFN is stable in water for at least 6-days (Fig. S3), it should be Fig. S5.

We will correct this.

Referee #3 (Comments on Novelty/Model System for Author):

Not so sure on quality of the drug due to broad range cysteine modifier

We recognise the point that the reviewer makes about SFX-01 modifying many protein cysteines as the reactive isothiocyanate within the compound is a simple molecule that may, at least at first glance, be anticipated to covalently adduct to a multitude of targets. However, 'covalent drugs' such as this are likely to primarily modify proteins that it first binds in the conventional (i.e., reversible) manner. This in some proteins provides a period of residence next to a cysteine thiol that gives sufficient time for the adduction chemistry. Put another way, SFX-01 does not react with thiols it fleetingly encounters, and the net outcome of this is that there is selectivity in the number of targets it reacts with and this in turn limits toxicity. The Chouchani lab reported in Cell 2022 (PMID: 32109415) that mice have at least ~171,000 redox modulated cysteines, whereas we found 47 proteins to be potentially modified by SFX-01. This is consistent with SFX-01 being selective in the

targets it modifies, as opposed to indiscriminately adducting all manner of redox regulated cysteines. This would explain why the drug, as outlined below, passed safety trials in animals and humans that culminated in clinical trials – with more in the pipeline.

On a final note, we acknowledge in the manuscript that poly-pharmacology likely contributes to the actions of SFX-01, albeit with Shp2 inhibition being a significant mechanism. Poly-pharmacology is likely a common feature of many drugs that are used clinically (see replies to Reveiwer1), and drugs with such multiple modes of action may be more effective against cancers.

Referee #3 (Remarks for Author):

The study "SFX-01 is therapeutic against myeloproliferative disorders caused by activating mutations in Shp2" is in format of a research article submitted to EMBO MOL MED. As such the work should contain experimental details that can be followed by the reader or reviewer and a clear description for Material and Methods section is available in the Supplementary Data, but more essential information should be moved to main text body since some of it is essential information. Critical data or information is not disclosed or introduced.

We submitted the manuscript in a 'generic' format, but in the revision will fully comply with the journal's requirements. We understand that the Reviewer wants more information, which is now provided within a reasonable text length. We note that the other two Reviewers did not mention insufficient detail and realise that some readers prefer a 'lighter style' because there is so much literature to digest nowadays.

The study has some interesting aspects and it focuses on orphan diseases at least with NS that do not have targeted drugs and maybe a dominance of childhood diseases that are not in interest of pharma industry, so these points are clearly positive and deserve credit. However, several shortcomings of the study are also noticeable; Drug specificity issues or suitability or off-target effects, toxicity issues will exist and several mechanistic things are quite limited to purified molecules and not much broadly applicable to proteomic data on cellular systems expressing SHP2 mutations relevant to Noonan Syndrome or MPN to be in main focus for targeting. As the manuscript was provided the reviewer suggests that it is not suited for publication in current format, but editing better and a number of experiments and controls can be performed for major revision to make it more round or better controlled for drug specificity issues on a broad range cysteine modifier and if that is really a good way to go. As it is, one cannot conclude on suitability of the drug in focused disease context or compared to other even less specific cysteine modifiers. If toxicity is in their model really not happening needs experimental validation and drug was not applied in dose-dependent evaluation, another bottleneck, but given in water ad libitum. Moreover, the text body and intro or result section lack detailed information as outlined in the following.

Below we outline our responses to the reviewer's points and describe how we will change the manuscript and what new experiments we propose performing to address their critique.

Major:

1) The discussion is in general quite limited and should more critically discuss known negative side effects and the possible applications so far of SFX-01 and it is not true that many clinical trials are done with it, only two are found completed on clinicaltrial.gov site from NIH which is USA government operated. Thus, the clinicaltrial.gov site is pretty comprehensive and only two studies listed with SFX-01 is limited and these are completed, one phase 2 in breast cancer as combi therapy, the other study very unrelated and irrelevant eventually, at least in regard to reviewer opinion.

We will change the manuscript text to describe more fully the clinical studies (past and on-going) with SFX-01. This is a link to a document that provides an independent overview of SFX-01 safety:

www.alzdiscovery.org/uploads/cognitive_vitality_media/SFX01-Cognitive-Vitality-For-Researchers.pdf

They conclude that SFX-01 has a “good safety profile, though limited number of studies. High doses are associated with gastrointestinal problems”.

Of course, before clinical trials there were safety tests in animals and in healthy humans that culminated in regulatory approval of clinical trials, which are on-going. We note that chronic human use of SFX-01 has not been reported but will capture this point as well in the revision.

Other than this several reports suggest it has broad range neurological disease action as a broad range acting drug due to covalent nucleophilic attack of cysteine residues. Here, it should be clearly introduced that all PTPs and not only SHP2 have catalytic Cysteine sites, so how about CD45 as the most expressed membrane tyrosine phosphatase in all blood cells except the erythroid lineage. Is CD45 also found to be pulled down in myeloid blasts? Can be a control blot and investigation. SFX-01 looks like a very unspecific broad range acting drug as a covalent cysteine modifier, maybe not as broad e.g. as "Stattic" a wrongly claimed STAT3 inhibitor, that might serve as a good model substance to compare it with, explained below in a separate point since that would allow a better conclusion for drug specificity issues and impact. At the end the authors claim that SFCX-01 is a good drug, but a comparison e.g. to Stattic which is a low-activity, non-peptide small molecule which turned out to be totally unspecific modifying any cysteine residue exposed on the surface of molecules could be a suitable assay if SFX-01 is more specific, then good, if not then the reviewer has strong doubts on toxicity and off-target effects. No toxicity testing in mice of their drinking water treatment was done, but this is mandatory. SFX-01 when dosed high was reported to have GI tract toxicity, but this was not analyzed, sections of GI tract, liver and kidney damage parameter, hematocytometry should be included, particularly due to placing the drug into drinking water which can be called uncontrolled or highly dosed? Mice can drink quite variable particularly if they have or have not disease, where NS model is quite harsh, but do the animals tolerate that compared to control group needs to be disclosed. Stattic was also published in a number of articles as a non selective STAT3 blocker or a broad range acting drug with questionable conclusion. The reviewer thinks the story would benefit on a comparison on some critical protein targets of SFX-01 and Stattic, SHP2, STAT1, STAT3 and STAT5B at least from expression and activity status and a couple of Western blots or proteomic could be runned to get an idea of a correct and state-of-the-art pharmacologic broad or more specific range cysteine modifier. Is SFX-01 really better than Stattic and if so why?

We hope the Reviewer is convinced by our arguments above and below that SFX-01 does not have the toxicity issues they might expect, perhaps because target engagement is less than might be initially imagined. And we are also mindful that SFX-01 has passed the rigorous regulatory approval process and now multiple studies have been and continue to be carried out. Whilst a comparison with Stattic would be interesting, we are unclear about the value of what this would add in terms of our focus on SFX-01 but know it we do not have the funds to perform it.

2) Authors should not only check on STAT1 and pYSTAT1 levels, they should also include and conclude on the analysis for the pY-STAT3 and pY-STAT5A/pYSTAT5B and total STAT3 as well as total STAT5A or STAT5B levels in their model cell systems, since covalent cysteine modifications are known actions on small molecular weight covalent binders of STAT molecules. STAT family members do also interact with SHP2 as it was also mapped in several papers and that can be cited. D type cyclins are in particular in myeloid or other acute leukemia or MPN targets of STAT3 and/or STAT5 action and they can be suppressed via STAT1 tumor suppressor induction. Thus, extended protein

assays should monitor STAT3, STAT5A and STAT5B expression and activity status, particularly since authors only focus on STAT1 action associated with interferon signaling largely. RNA-seq pathway analysis with and without treatment in diseased mouse context should better shed light on core cancer pathway activity.

The Reviewer suggested a series of extensive experiments in this paragraph on multiple STAT proteins in various model systems, together with RNA-seq in mice with or without SFX-01. This together with the studies above (i.e., proteomics) and below represents a huge undertaking that could exceed the amount of work presented in the current manuscript. The reality is that it would take years of work and funds that are not available to us. We have performed several studies that overall strengthen the evidence for the conclusions drawn from this project overall. This includes the extensive unbiased phosphoproteomics analysis that showed, as can be seen in the Table below and in Figure 5G, that significant increases in phospho-STAT1 at multiple sites occurred in response to SFX-01. No changes in STAT 3 or 5 phosphorylation in the same samples were detected.

	HET_Drug_vs_HET_			WT_Drug_vs_WT_		
	difference	pvalues	FDR	difference	pvalues	FDR
Stat1(S727);	-0.296646455	0.375944	0.648738	-0.48358	0.434178	0.687443
Stat1(S727);Stat1(M734);Stat1(M726);	0.832649978	0.00926	0.074511	0.278043	0.155971	0.433078
Stat1(M734);Stat1(T720);	1.480849712	0.000241	0.004799	1.544107	0.002291	0.033731
Stat1(T719);	-0.204015353	0.465796	0.72399	-0.40985	0.445749	0.697467
Stat1(T719);Stat1(M726);	2.107621226	0.000117	0.002586	1.440538	0.001914	0.029585
Stat1(M734);Stat1(T719);Stat1(M726);	0.783745292	0.007145	0.062848	0.289334	0.3867	0.650599
Stat3(181-197, no_mod);	-0.401102173	0.117674	0.35003	-0.44175	0.118011	0.384192
Stat3(M185);	0.686862745	0.108307	0.332435	0.272267	0.486454	0.724551
Stat5a(C126);Stat5a(S127);	0.230126391	0.185853	0.452875	-0.0685	0.80809	0.917833
Stat5a(S779);	0.251656378	0.316804	0.596885	0.263932	0.211374	0.489531

3) RNA-seq analysis with and without drug would allow a better conclusion on pathway analysis hit by SFX-01 versus vehicle control in drinking water of mice of isolated myeloid blasts that are e.g. FACS sorted and purified and sequenced in bulk as a good enough analysis in three experimental mice each group to conclude on targeting aspects. Here, a good surface molecule target of STAT1 action should be MHC class I surface expression which can be monitored in parallel as well. The consequence of it might be foreign neoantigen expression, but that could be discussed if changed. Thus, authors could look by FACS on HLA subtype expression level on their mice which are most likely inbred pure background strains, which is another important information for readers to be disclosed. MHC class I should be upregulated upon their drug treatment. Other direct STAT1 targets are caspase induction or p21 cell cycle inhibitor induction or interferon regulatory factor induction or antiviral genes, where Mx1 GTPase are good targets to be e.g. eluted or to more broadly see and discuss interferon target genes. Unbiased RNA-seq would reveal better insight. Thus, a more broader anti-survival, anti-proliferative or anti-inflammatory action could originate from a broader impact on JAK-STAT-SOCS protein components, known to be essential genes in MPN and acute leukemias of the myeloid lineages.

Again, to fulfil what is suggested by the Reviewer here would be a huge amount of additional work that does not necessarily directly connect with what we have currently trying to report and conclude. Essentially the Reviewer suggests an entire programme of work, which we consider to be beyond the scope of this study. However, we will perform the phospho-proteomics studies outlined above.

4) Normally text editing is minor, but since quite a lot of info in this manuscript is missing the reviewer opens here another point as major for lack of specific information to better grasp the study or to understand the nature of the model, the expression, etc....As the manuscript was written one does not have the impression that details might be so clear why e.g. cardiac tissue from these mice was used for analysis when the headline states "myeloproliferative disorder", but it is clear when reading up on these mice that the cardiac tissue is affected in half of the mice and how the transgenic model was generated should be stated as well in detail, e.g. is it a knockin, is it a transgene that is randomly integrated, that matters at the end to understand the model, it expresses one SHP2 variant found in Noonan syndrome, etc.... Thus, advice is to write that clearly in with own words into the introduction or into the result section even for mouse model description, essential information is otherwise lost. Thus, the reviewer feels that the mouse model info is crucial for judgment and it makes the reviewing process faster if such information would be provided, since the reader/reviewer has then not to look that up and also most readers will not understand that based on background of the wealth of mouse models available today and differences in model character for expression, tissue type expression, activation status due to mutation, etc. The description by the Ben Neel lab on the Noonan mouse model carrying the SHP2 variant is very detailed in the abstract and why not using that take home info into the manuscript. Here, it is quoted from the abstract in following lines: "... mice expressing the Noonan syndrome-associated mutant D61G. When homozygous, the D61G mutant is embryonic lethal, whereas heterozygotes have decreased viability. Surviving Ptpn11D61G/+ embryos (~50%) have short stature, craniofacial abnormalities similar to those in Noonan syndrome, and myeloproliferative disease. Severely affected Ptpn11D61G/+ embryos (~50%) have multiple cardiac defects similar to those in mice lacking the Ras-GAP protein neurofibromin. Their endocardial cushions have increased Erk activation, but Erk hyperactivation is cell and pathway specific....."

Greater experimental details, including mouse genetic information, are be provided in the revised manuscript, but we tried to keep the text length palatable to the average reader.

Minor:

5) Along the lines on missing information also the structure and Mw, etc. should be given as well as the R- and L-forms for enantiomer should be disclosed. Evgen Pharma data disclosed that MW: 173.19 + 972.846 g/mol is within suitable range of Lipinski Rule of 5 (<http://www.scfbio-iitd.res.in/software/drugdesign/lipinski.jsp>), which could be discussed as well by the authors to claim it has drug like properties that are good.

We have included the information requested and text explain Lipinski's rules are met by SFX-01.

8th Apr 2025

Dear Prof. Eaton,

Thank you for submitting your manuscript to EMBO Molecular Medicine. We have now received the reports from referees #1 and #2.

As you will see below, while reviewer #2 is satisfied with the revisions, reviewer #1 still has some concerns. This reviewer also evaluated your responses to reviewer #3 and mentioned that most of this reviewer's requests have been excluded, except for the toxicity assay. Reviewer #3 also commented that RNAseq experiments would have been an interesting complementary approach to characterize the effects of SFX-01.

After discussion within the team, we would like to invite further revisions of the manuscript to address any remaining concerns with experimental data where possible or adequate discussion. Please note that RNAseq experiments will NOT be required for further consideration.

As EMBO Press usually encourages one single round of revisions, please be aware that this will be the last chance for you to address the reviewers' concerns. The revised manuscript will once again be subjected to review, and we cannot guarantee a positive outcome at this stage.

If you would like to discuss further the points raised by the reviewers, I am available to do so via email or video. Let me know if you are interested in this option.

Moreover, please address the following editorial requests:

1. Please provide up to 5 keywords.
 2. Author contributions: CRediT has replaced the traditional author contributions section because it offers a systematic machine readable author contributions format that allows for more effective research assessment. Please remove the Authors Contributions from the manuscript and use the free text boxes beneath each contributing author's name in our system to add specific details on the author's contribution. More information is available in our guide to authors.
 3. Please rename the conflict-of-interest section to "Disclosure and competing interests statement".
 4. References should be listed alphabetically, with 10 author names before et al.
 5. We replaced Supplementary Information with Expanded View (EV) Figures and Tables that are collapsible/expandable online. EV Figures should be cited as 'Figure EV1, Figure EV2" etc... in the text and their respective legends should be included in the main text after the legends of regular figures.
 - For the figures that you do NOT wish to display as Expanded View figures, they should be bundled together with their legends in a single PDF file called *Appendix*, which should start with a short Table of Content. Appendix figures should be referred to in the main text as: "Appendix Figure S1, Appendix Figure S2" etc.
 - Additional Tables/Datasets should be labelled and referred to as Table EV1, Dataset EV1, etc. Legends must be provided in a separate tab in case of .xls files. Alternatively, the legend can be supplied as a separate text file (README) and zipped together with the Table/Dataset file.
- See detailed instructions here:

6. Supplementary methods should be moved to the main manuscript text.
7. Source Data should be uploaded as one zipped file per figure. Please rename "Appendix table 1" to Dataset EV1, add a legend and a short description.
8. Please upload a reagent table (using our template).
9. Synopsis: please resize to 550x300-600 pixels and remove the synopsis text from the manuscript file and upload as a separate file.
10. Figures and figure panels should all be referenced in the text, in chronological order (currently, Appendix Figure S8 is called out before S7, and S13 before S12).
11. Please remove "materials and correspondence" from the manuscript text.
12. Please address the queries from our copy editors in the figure legends:
 - Please note that the exact p values are not provided in the legends of figures 2C, 3A, 4B, 5A
 - Please indicate what */ **/ ***/ **** represents; if this represents p value(s), please specify the exact p value in the legend(s) of figure(s) 4D
 - Please indicate what */ **/ ***/ **** represents; if this represents p value(s), please specify the test and the exact p value in the legend(s) of figure(s) 4F
 - Please note that information related to n is missing in the legends of figure 1E
 - Please note that n=2 in figure 5D. Please use scatter blots showing the individual datapoints in these cases. The use of statistical tests needs to be justified.
 - Please note that the error bars are not defined in the legends of figures 5C-F.

As part of the EMBO Publications transparent editorial process initiative (see our Editorial at <http://embomolmed.embopress.org/content/2/9/329>), EMBO Molecular Medicine will publish online a Review Process File (RPF) to accompany accepted manuscripts.

In the event of acceptance, this file will be published in conjunction with your paper and will include the anonymous referee reports, your point-by-point response and all pertinent correspondence relating to the manuscript. Let us know whether you agree with the publication of the RPF and as here, if you want to remove or not any figures from it prior to publication. Please note that the Authors checklist will be published at the end of the RPF.

I look forward to receiving your revised manuscript.

Sincerely,

Lise Roth

**** Reviewer's comments ****

Referee #1 (Remarks for Author):

In this revised version, Hyun-Ju Cho and coworkers have significantly improved their initial manuscript by addressing some of the reviewers' issues. They provide more evidence to support causality (Fig S14). To address reviewers' concerns about off target effects, they performed toxicity assay, revealing no major outcomes in accordance with previous studies, and developmental assay (Fig S6, S9). They also provide new phosphoproteomics data highlighting phosphorylation changes upon SFX-01 treatment in myeloid cells (Fig 5).

Below are my comments

- in the toxicity assay, it is interesting to note that WT, but not NS mice, put on weight upon SFX-01 treatment, any hypothesis about this? Moreover, patients with NS often show low cholesterol, which has been also reported in the NS mice (D61G/+), but not in this study. Have the animals been starved before blood collection?
- NS mice phenotyping upon SFX-01 treatment: in response to my suggestion to document additional features, the authors state NS mice do not develop pathogenic features at adulthood based on the first description of the NS model from B. Neel' lab. However, since 2004, a few publications allowed to refine these initial observations. For instance, cardiac dysfunction in NS D61G can be targeted postnatally with dasatinib (PMID: 27942593). Moreover, the very same NS mouse model has increased bleeding time in relation with defective aggregation (PMID: 32526025). Therefore, the sentence p12 "For example.... platelet aggregation" must be removed. Of interest for the authors, in this paper, they showed reduced Syk phosphorylation (although on different residues) in platelets from NS mice, which could be consistent with the phosphoproteomics showed in this manuscript. Growth retardation is also an interesting feature that could have been tested in young NS mice. Instead of those, the authors choose to assess the impact of SFX-01 exposure during pregnancy, that results in loss of heterozygous offsprings. This may reflect an aggravation of the phenotype (maybe in relation with increased ERK phosphorylation during development?), or highlight significant side effect. In Fig S9, which of the parents carry the mutation in the HETxWT breeding? FigS9D is a bit weird, for the HETxHET condition, considering there is no litter.
- throughout the manuscript, the authors imply that SFX-01 is an inhibitor of activating mutants of SHP2 (p 2, 4, 5, 7, 15/16...), which to my opinion is an overstatement, as the authors do not provide evidence for SFX-01 being more efficient on mutated SHP2 than on WT SHP2. SFX-01 inhibits both WT and mutated SHP2 (Fig 2, S14), which does not call into question its therapeutic effect in NS mice or in JMML, and the phosphoproteomic interpretation of SHP2 Y580 dephosphorylation to support selectivity is a bit speculative. Experiments (e.g. differential SFX-01 dependent inhibition in vitro on WT vs mutated SHP2) must be provided to demonstrate the selectivity of SFX-01 for mutated vs WT SHP2 or, if not, the text must be modified accordingly.
- the phosphoproteomics analysis is an important body of data. However, several of the captured phosphoproteins cannot be direct SHP2 targets as being not phosphorylated on tyrosine. Conclusions from this analysis should maybe be dampened as reduced proliferation would probably result in overlapping pattern. Few explanations regarding the phosphoproteomics analysis are also required for the reader's understanding. What does the third column in the heat map describe? NS vs WT with or without SFX-01?

Minor: title of figure S14: SHP2 should read SHP099. This figure may deserve being in a main figure.

Referee #2 (Comments on Novelty/Model System for Author):

The authors have addressed the issues and concerns raised, I'm satisfied with the revision.

Referee #2 (Remarks for Author):

The authors have done an excellent job in revising the manuscript, and have addressed the issues and concerns raised very well.

Referee #1 (Remarks for Author):

In this revised version, Hyun-Ju Cho and coworkers have significantly improved their initial manuscript by addressing some of the reviewers' issues. They provide more evidence to support causality (Fig S14). To address reviewers' concerns about off target effects, they performed toxicity assay, revealing no major outcomes in accordance with previous studies, and developmental assay (Fig S6, S9). They also provide new phosphoproteomics data highlighting phosphorylation changes upon SFX-01 treatment in myeloid cells (Fig 5).

Below are my comments

We thank the Reviewer for giving their valuable time to reviewing our study.

- in the toxicity assay, it is interesting to note that WT, but not NS mice, put on weight upon SFX-01 treatment, any hypothesis about this?

This is an interesting observation. We thought that there wasn't any significant growth in WT mice given SFX-01 and that the separation with the two WT growth curves was primarily due to the treatment group starting slightly higher in mass than the control group at time 0. However, we have now assessed these data with statistics and found that indeed SFX-01 increases the rate of growth of WTs. We have added some text to the revision to highlight this difference and included some discussion addressing the Reviewers question about why this might be. Indeed, in addressing this point we noted that SFX-01 treatment significantly increased food intake by 2.5 g/week compared to controls ($P = 0.0051$) and have added text to say this approximate 10% increase in calories likely explains the additional weight gain. We speculate that this may be connected with sulforaphane from SFX-01 altering the gut microbiome to influence food intake (PMID: 33338213), although we note that the electrophile did not have any adverse effect on the small intestine of either genotype (Appendix Figure S6). It is notable that Noonan syndrome is associated with feeding difficulties, gut dysmotility and behavioural disorders causing avoidant or restrictive intake of food (PMID: 10373129, PMID: 770238). However, it remains unclear why SFX-01 increased food consumption and weight gain in WT and why this did not occur in the transgenics.

- Moreover, patients with NS often show low cholesterol, which has been also reported in the NS mice (D61G/+), but not in this study. Have the animals been starved before blood collection?

We did not starve the mice before the collection of blood for the various biochemical measurements made, including cholesterol. Perhaps, this explains the difference between this and the other studies mentioned by the Reviewer.

- NS mice phenotyping upon SFX-01 treatment: in response to my suggestion to document additional features, the authors state NS mice do not develop pathogenic features at adulthood based on the first description of the NS model from B. Neel' lab. However, since 2004, a few publications allowed to refine these initial observations. For instance, cardiac dysfunction in NS D61G can be targeted postnatally with dasatinib (PMID: 27942593). Moreover, the very same NS mouse model has increased bleeding time in relation with defective aggregation (PMID: 32526025). Therefore, the sentence p12 "For example.... platelet aggregation" must be removed. Of interest for the authors, in

this paper, they showed reduced Syk phosphorylation (although on different residues) in platelets from NS mice, which could be consistent with the phosphoproteomics showed in this manuscript.

We have revised the text as suggested to reflect the Reviewer's point that PMID: 27942593 and PMID: 32526025 provided evidence after the initial Neel lab publication that the transgenics have cardiac dysfunction and defective platelet aggregation respectively. We are grateful to the reviewer for also highlighting the connection with Syk phosphorylation and added some discussion text to reflect this point and the rational potential of exploring the impact of SFX-01 on defective platelet aggregation in Noonan syndrome.

- Growth retardation is also an interesting feature that could have been tested in young NS mice.

We did not perform these studies, having to be selective in those we prioritised. There was no increase in growth rate in older mice upon SFX-01 treatment, but a study in younger mice would have been interesting.

Instead of those, the authors choose to assess the impact of SFX-01 exposure during pregnancy, that results in loss of heterozygous offsprings. This may reflect an aggravation of the phenotype (maybe in relation with increased ERK phosphorylation during development?), or highlight significant side effect. In Fig S9, which of the parents carry the mutation in the HETxWT breeding? FigS9D is a bit weird, for the HETxHET condition, considering there is no litter.

We agree with the Reviewer that the loss of offspring may be a result of increased pERK activation induced by SFX-01 that occurs in mice or cells exposed to this drug (Appendix Figure S12D-F) observed both by immunoblotting for pERK and with unbiased phosphoproteomics analysis. Indeed, we have now included additional pERK data from neonates born from WTxWT crosses in which the mother was exposed to vehicle or SFX-01 (see Appendix Figure S9I). This again shows the drug increases pERK, and we have now included discussion text citing various papers that report activation of this kinase *in utero* is harmful and causes cardiac malformations.

We agree with the Reviewer that it is difficult to understand why administering SFX-01 to mothers carrying offspring from HETxHET crosses deliver no pups. This complete loss of offspring only occurred at the 2.5 mg/ml SFX-01 dose (Appendix Figure S9D), whereas some WT offspring were delivered when the lower 0.8 mg/ml dose was used (Appendix Figure S9H). Thus, SFX-01-induced loss of transgenic mice may cause stress that indirectly results in loss of WT mice that otherwise would be born. Indeed, WTxWT crosses exposed to SFX-01 *in utero* does not result in a decrease in litter size and so the losses are connected with carrying the mutant Ptpn11 gene. However, there is complexity because the loss of mice may relate to the genotype of the mother, as the losses only occur when she has the HET genotype and not WT and this could be connected with the syndromic mice being smaller and their capacity to carry litters especially in the setting of any additional stress caused by pERK activation by SFX-01. Overall, a potential explanation for the 2.5 mg/ml dose of SFX-01 causing embryonic lethality may be because of potentiated pERK, but as no pups were delivered this could not be assessed.

- throughout the manuscript, the authors imply that SFX-01 is an inhibitor of activating mutants of SHP2 (p 2, 4, 5, 7, 15/16...), which to my opinion is an overstatement, as the authors do not provide evidence for SFX-01 being more efficient on mutated SHP2 than on WT SHP2. SFX-01 inhibits both WT and mutated SHP2 (Fig 2, S14), which does not call into question its therapeutic effect in NS mice or in JMML, and the phosphoproteomic interpretation of SHP2 Y580 dephosphorylation to support selectivity is a bit speculative.

We understand the Reviewer's point as we did repeatedly speculate that the open conformation mutant should be more susceptible to SFX-01 modification due to enhanced access to and increased reactivity of the catalytic cysteinyl thiol. We thought that we had not overstated this point but realise now that we need to be more cautious. Therefore, as we have not provided empirical evidence that mutant Shp2 is more susceptible to modification by the electrophile, we have gone through the manuscript and toned down this speculation and included new text reiterating this limitation.

- Experiments (e.g. differential SFX-01 dependent inhibition in vitro on WT vs mutated SHP2) must be provided to demonstrate the selectivity of SFX-01 for mutated vs WT SHP2 or, if not, the text must be modified accordingly.

We understand that we have not provided direct experimental evidence that D61E Shp2 is more sensitive to oxidative / adductive inhibition by SFX-01 compared to WT. We have been trying to do this for a long time as we understand it's value, consistent with the reviewer's previous comment and this one. We quite simply could not make soluble, functional recombinant D61E Shp2 as we have for other enzymes we study and nor could our core university Protein Production Facility – perhaps consistent with it not being commercially available. We tried transfection and immunoprecipitation studies but perhaps because the lability of the adduct and transition to the dithiolethione and the time taken to immunocapture the Shp2, these studies failed. As per the guidance of the reviewer we have modified the text accordingly to acknowledge this limitation. Indeed, we now include text saying “the D61G variant is potentially more susceptible but we lack empirical evidence to support this notion”. We reiterate this later in the text by saying “Although this may be because the D61G mutation increases the nucleophilicity of the Shp2 catalytic cysteine or access to it because of altered hydrogen bonding⁴⁹, experimental evidence for this conjecture is lacking”.

- the phosphoproteomics analysis is an important body of data. However, several of the captured phosphoproteins cannot be direct SHP2 targets as being not phosphorylated on tyrosine. Conclusions from this analysis should maybe be dampened as reduced proliferation would probably result in overlapping pattern.

Whilst there are studies concluding Shp2 can dephosphorylate PKR at Thr446 (PMID: 20872791) and others showing PThr/pSer status can influence proximal pTyr dephosphorylation by Shp2 (PMID: 36114179), we agree the changes in phosphorylation on non-Tyr residues is likely due to events secondary or independent of the phosphatase being inhibited. We have noted this by additions to the revised manuscript but have also included text considering that every protein mentioned in the heat map in Figure 5G is a substrate of Shp2 or is known to interact with it. Thus, it is notable that proteins closely connected with Shp2 undergo altered phosphorylation at Tyr when cells are exposed to SFX-01, but also, perhaps unexpectedly at Thr and Ser.

- Few explanations regarding the phosphoproteomics analysis are also required for the reader's understanding. What does the third column in the heat map describe? NS vs WT with or without SFX-01?

We have included additional text within the main manuscript about the phosphoproteomics experiments as advised. We have now also explained in the legend for Figure 5G that this comparison relates to relative levels of those phosphorylation sites in the KI transgenic mouse compared to WT under basal conditions.

Minor: title of figure S14: SHP2 should read SHP099. This figure may deserve being in a main figure.

We have corrected the title and moved this, so it is now included as an Expanded View figure (EV1) in the manuscript as suggested.

Referee #2 (Comments on Novelty/Model System for Author):

The authors have addressed the issues and concerns raised, I'm satisfied with the revision.

Referee #2 (Remarks for Author):

The authors have done an excellent job in revising the manuscript, and have addressed the issues and concerns raised very well.

We thank the Reviewer for giving their valuable time to reviewing our study.

6th Jun 2025

Dear Prof. Eaton,

Thank you for submitting your manuscript to EMBO Molecular Medicine. Referee #1 has reviewed your revised manuscript and is now supportive of publication. I will therefore be able to accept your manuscript once the following editorial concerns are addressed:

1/ Manuscript text:

- Given the number of figures, appendix and size of the current manuscript, the "Report" format is not suitable and the piece should follow an "Article" type. This does not affect the general formatting of the manuscript; however, the results and discussion should be two individual sections.
- Please remove the coloured font text and only keep in track changes mode any new modification.
- "Materials and Methods" should be renamed "Methods":
 - o Cells: please indicate whether the cells were authenticated and tested for mycoplasma contamination.
 - o Antibodies: please provide dilutions/concentrations for all antibodies.
 - o Animals: please provide the reference number for ethics approval.
 - o Patient samples: please include the full statement that the experiments conformed to the principles set out in the WMA Declaration of Helsinki and that the experiments conformed to the principles set out in the WMA Declaration of Helsinki and the Department of Health and Human Services Belmont Report.
 - o Statistics: please provide a statement on inclusion/exclusion criteria.
- Data availability: please note that all data must be public before acceptance of the manuscript.
- The Figure Legends should be placed after the References, at the very end of the manuscript.

2/ Source Data:

Please provide the completed source data checklist. Please provide Source data for Figure 1B, Figure 4A, B (Western Blots). Please check the labeling of your Source Data (i.e Fig. 5A, B).

3/ Checklist:

- you indicated "Yes" in the category "Plants", please double check.
- please fill out the entire section on "Experimental study design and statistics"
- in several instances, you indicated that the information was provided in the "Expanded View Content", please correct or clarify.

4/ Thank you for providing a synopsis text and image. Please add a stand-first to the text (maximum 300 characters, including space). I have cropped the attached portion of your synopsis to serve as thumbnail on our website, please let me know if you agree, or provide another image at the right dimensions (115x70 pixels).

5/ As part of the EMBO Publications transparent editorial process initiative (see our Editorial at <http://embomolmed.embopress.org/content/2/9/329>), EMBO Molecular Medicine will publish online a Review Process File (RPF) to accompany accepted manuscripts.

This file will be published in conjunction with your paper and will include the anonymous referee reports, your point-by-point response and all pertinent correspondence relating to the manuscript. Let us know whether you agree with the publication of the RPF and as here, if you want to remove or not any figures from it prior to publication.

I look forward to receiving your revised manuscript.

With kind regards,

Lise Roth

**** Reviewer's comments ****

Referee #1 (Remarks for Author):

The authors addressed my different concerns in this new version of their manuscript.

The authors addressed the remaining editorial issues.

24th Jun 2025

Dear Prof. Eaton,

I am pleased to inform you that your manuscript is accepted for publication and is now being sent to our publisher to be included in the next available issue of EMBO Molecular Medicine!

With kind regards,

Lise Roth
